# Meta-DT: Offline Meta-RL as Conditional Sequence Modeling with World Model Disentanglement

**Zhi Wang**[1]  **Li Zhang**[1]  **Wenhao Wu**[1]  **Yuanheng Zhu**[2]  **Dongbin Zhao**[2]  **Chunlin Chen**[1*]

[1]Nanjing University    [2]Institution of Automation, Chinese Academy of Sciences

{zhiwang, clchen}@nju.edu.cn    {lizhang, whao_wu}@smail.nju.edu.cn

{yuanheng.zhu, dongbin.zhao}@ia.ac.cn

## Abstract

A longstanding goal of artificial general intelligence is highly capable generalists that can learn from diverse experiences and generalize to unseen tasks. The language and vision communities have seen remarkable progress toward this trend by scaling up transformer-based models trained on massive datasets, while reinforcement learning (RL) agents still suffer from poor generalization capacity under such paradigms. To tackle this challenge, we propose Meta Decision Transformer (Meta-DT), which leverages the sequential modeling ability of the transformer architecture and robust task representation learning via world model disentanglement to achieve efficient generalization in offline meta-RL. We pretrain a context-aware world model to learn a compact task representation, and inject it as a contextual condition to the causal transformer to guide task-oriented sequence generation. Then, we subtly utilize history trajectories generated by the meta-policy as a self-guided prompt to exploit the architectural inductive bias. We select the trajectory segment that yields the largest prediction error on the pretrained world model to construct the prompt, aiming to encode task-specific information complementary to the world model maximally. Notably, the proposed framework eliminates the requirement of any expert demonstration or domain knowledge at test time. Experimental results on MuJoCo and Meta-World benchmarks across various dataset types show that Meta-DT exhibits superior few and zero-shot generalization capacity compared to strong baselines while being more practical with fewer prerequisites. Our code is available at https://github.com/NJU-RL/Meta-DT.

## 1 Introduction

Building generalist models that solve various tasks by training on vast task-agnostic datasets has emerged as a dominant paradigm in the language [1] and vision [2] communities. Offline reinforcement learning (RL) [3] allows learning an optimal policy from trajectories collected by some behavior policies without access to the environments, which holds tremendous promise for turning massive datasets into powerful generic decision-making engines [4, 5, 6], akin to the rise of large language models (LLMs) like GPT [7]. A performant example is the decision transformer (DT) [8] that leverages the transformer's strong sequence modeling capability for trajectory data and achieves promising results on offline RL [9, 10, 11, 12].

Transformer-based large models have shown remarkable scaling properties and high transferability across various domains, including language modeling [13, 7], computer vision [14, 15], and image generation [16, 17]. However, their counterparts in RL are usually specialized to a narrowly defined task and struggle with poor generalization to unseen tasks, due to distribution shift and the lack of

---

*Correspondence to Chunlin Chen <clchen@nju.edu.cn>.

self-supervised pretraining techniques [18, 19, 20]. To promote generalization, Prompt-DT [21] uses expert demonstrations as a high-quality prompt to encode task-specific information. Generalized DT [22] conditions on some hindsight information, e.g., statistics of the reward distribution, to match task-specific feature distributions. These works usually rely on domain knowledge at test time, e.g., expert demonstrations or hindsight statistics, which is expensive and even infeasible to acquire in advance for unseen tasks [23]. The aforementioned limitations raise a key question: *Can we design an offline meta-RL framework to achieve efficient generalization to unseen tasks while drawing upon advances in the sequence modeling paradigm with the scalable transformer architecture?*

In offline RL, the collected dataset depends on the task and behavior policy. When behavior policies are highly correlated with tasks in the training dataset, the agent tends to memorize the features of behavior policies and produce biased task inference at test time due to the change of behavior policies [24, 25]. One major challenge for generalization is how to accurately encode task-relevant information for extrapolating knowledge across tasks, while minimizing the requirement on the distribution of pre-collected data and behavior policies. RL agents typically learn through active interaction with the environment, and the transition dynamics $p(r, s'|s, a)$ completely describes the characteristics of the underlying environment [26]. The environment dynamics, also called *world model* [27, 28], is intrinsically invariant to behavior policies or collected datasets, thus presenting a promising alternative for robustly encoding task beliefs. By capturing compact representations of the environment, world models carry the potential for substantial transfer between tasks [29], continual learning [30], and generalization from offline datasets [31, 32].

Inspired by this, we propose a novel framework for offline meta-RL, named Meta-DT that leverages robust task representation learning via world model disentanglement to conduct task-oriented sequence generation for efficient generalization. First, we pretrain a context-aware world model to capture task-relevant information from the multi-task offline dataset. The world model contains an encoder that abstracts dynamics-specific information into a compact task representation, and a decoder that predicts the reward and state transition functions conditioned on that representation. Second, we inject the task representation as a contextual label to the transformer to guide task-conditioned sequence generation. In this way, the autoregressive model learns to estimate the conditional output of multi-task distributions and achieve desired returns based on the task-oriented context. Finally, we leverage the past trajectories generated by the meta-policy as a self-guided prompt to exploit the architecture inductive bias, akin to Prompt-DT [21]. We feed available trajectory segments to the pretrained world model and choose the segment with the largest prediction error to construct the prompt, aiming to encode task-specific information *complementary* to the world model maximally.

In summary, our main contributions are as follows:

- **Generalization Ability.** We propose Meta-DT to leverage the conditional sequence modeling paradigm with the scalable transformer architecture to achieve efficient generalization across unseen tasks without any expert demonstration or domain knowledge at test time.

- **Context-Aware World Model.** We introduce a context-aware world model to learn a compact task representation capable of generalizing across a distribution of varying environment dynamics.

- **Complementary Prompt.** We design a self-guided prompt that encodes task-specific information complementary to the world model maximally, harnessing the architectural inductive bias.

- **Superior Performance.** Experiments on various benchmarks show that Meta-DT exhibits higher few and zero-shot generalization capacity, while being more practical with fewer prerequisites.

## 2  Related Work

**Offline Meta-RL.** Offline RL [33] allows learning optimal policies from pre-collected offline datasets without online interactions with the environment [34, 35]. Offline meta-RL learns to generalize to new tasks via training on a distribution of such offline tasks [25, 36]. Optimization-based meta-learning methods [37, 38] seek policy parameter initialization that requires only a few adaptation steps to new tasks. MACAW [39] introduces this architecture into value-based RL subroutines, and uses simple supervised regression objectives for sample-efficient offline meta-training. On the other hand, context-based meta-learning methods learn a context encoder to perform approximate inference on task representations, and condition the meta-policy on the approximate belief for generalization, such as PEARL [40], VariBAD [41], FOCAL [42], CORRO [24], CSRO [25], and UNICORN [43]. These works extended from the online meta-RL setting are mainly trained by temporal difference

learning, the dominant paradigm in RL [9, 44]. This paradigm might be prone to instabilities due to function approximation, off-policy learning, and bootstrapping, together known as the deadly triad [26]. Moreover, many of these works rely on hand-designed heuristics to keep the policy within the offline dataset distribution [45]. This motivates us to turn to harness the lens of conditional sequence modeling with the transformer architecture to scale existing offline meta-RL algorithms.

**RL as Sequence Modeling.** The recent rapid progress in (self) supervised learning models is in large part predicted by empirical scaling laws with the transformer architecture, such as GPT [7], ViT [14], and DiT [46]. Decision transformer [8] first introduces the transformer's sequence modeling capacity to solving offline RL without temporal difference learning, which autoregressively outputs optimal actions conditioned on desired returns. As a concurrent study, trajectory transformer [47] directly models distributions over sequences of states, actions, and rewards, followed by beam search as a planning algorithm in a model-based manner. Extending the LLM-like structure to RL, this paradigm activates a new pathway toward scaling powerful RL engines with large-scale compute and data [48].

For multi-task learning, Gato [49] trains a multi-modal generalist policy in the GPT style with a decoder-only transformer. Multi-game DT [4] trains a suite of up to 46 Atari games simultaneously, and controls policy generation at inference time with a pretrained classifier that indicates the expert level of behaviors. For generalization to unseen tasks, Prompt-DT [21] exploits a prefix prompt architecture and Generalized DT [22] designs a hindsight information matching structure to encoder task-specific information. These works usually require domain knowledge at test time, such as expert demonstrations or hindsight statistics. T$^3$GDT [50] conditions action generation on three-tier guidance of global transitions, local relevant experiences, and vectorized time embedding. CMT [18] provides a pretrain and prompt-tuning paradigm where a task prompt is extracted from offline trajectories with an encoder-only transformer to realize few-shot generalization.

## 3 Preliminaries

### 3.1 Offline Meta Reinforcement Learning

RL is generally formulated as a Markov decision process (MDP) $M = \langle \mathcal{S}, \mathcal{A}, T, R, \gamma \rangle$, where $\mathcal{S}/\mathcal{A}$ is the state/action space, $T/R$ is the state transition/reward function, and $\gamma$ is the discount. The objective is to find a policy $\pi(a|s)$ to maximize the expected return as $\max_\pi J_M(\pi) = \mathbb{E}_\pi \left[ \sum_{t=0}^\infty \gamma^t R(s_t, a_t) \right]$.

In offline meta-RL, we assume that the task follows a distribution $M_i = \langle \mathcal{S}, \mathcal{A}, T_i, R_i, \gamma \rangle \sim P(M)$, where tasks share the same state-action spaces while vary in the reward and state transition functions, i.e., environment dynamics. For each task $i$ from a total of $N$ training tasks $\{M_i\}_{i=1}^N$, an offline dataset $\mathcal{D}_i = \{(s_{i,j}, a_{i,j}, r_{i,j}, s'_{i,j})\}_{j=1}^J$ is collected by an arbitrary behavior policy $\pi_\beta^i$. The agent can only access the offline datasets $\{\mathcal{D}_i\}_{i=1}^N$ to train a meta-policy $\pi_{\text{meta}}$. At test time with the *few-shot* setting, given an unseen task $M \sim P(M)$, the agent can access a small dataset $\mathcal{D} = \{(s_j, a_j, r_j, s'_j)\}_{j=1}^{J'}$ to construct the task prompt or infer the task belief before policy evaluation. For the *zero-shot* setting, the trained meta-policy is directly evaluated on the unseen task by interacting with the test environment to estimate the expected return. The objective is to learn a meta-policy to maximize the expected episodic return over test tasks as $J(\pi_{\text{meta}}) = \mathbb{E}_{M \sim P(M)} \left[ J_M(\pi_{\text{meta}}) \right]$.

### 3.2 Decision Transformer

By leveraging the superiority of the attention mechanism [13] in extracting dependencies between sequences, transformers have gained substantial popularity in the language and vision communities [7, 14]. Decision transformer [8] introduces associated advances to solve RL problems and re-frames offline RL as a return-conditioned sequential modeling problem, inheriting the transformer's efficiency and scalability when modeling long trajectory data. DT models the probability of the next sequence token $x_t$ conditioned on all tokens prior to it: $P_\theta(x_t|x_{<t})$, similar to contemporary decoder-only sequence models [51]. The sequence we consider has the form: $x = (\cdots, \hat{R}_t, s_t, a_t, \cdots)$, where $\hat{R}_t$ is the agent's target return for the rest of the trajectory. Such a sequence order respects the causal structure of the decision process. When training with offline data, $\hat{R}_t = \sum_{i=t}^T r_i$, and during testing, $\hat{R}_t = G^* - \sum_{i=0}^t r_i$, where where $G^*$ is the target return for an entire episode. The same timestep embedding is concatenated to the embeddings of $\hat{R}_t$, $s_t$, and $a_t$, and the head corresponding to the state token is trained to predict the action by minimizing its error from the ground truth.

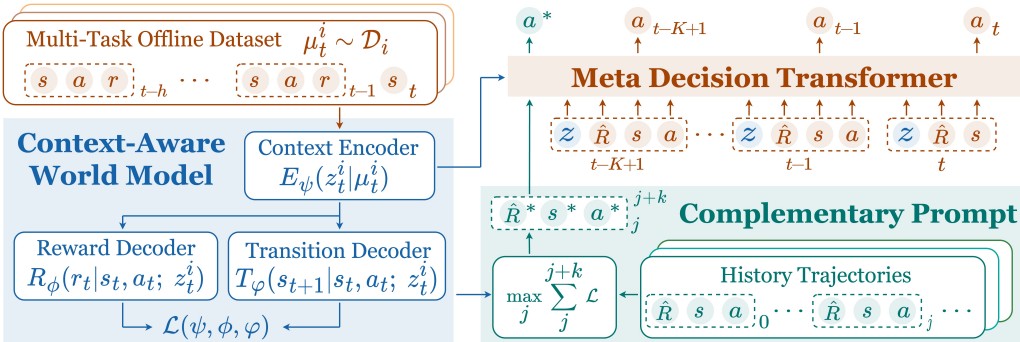

Figure 1: The overview of Meta-DT. We pretrain a context-aware world model to accurately disentangle task-specific information. It contains a context encoder $E_\psi$ that abstracts recent $h$-step history $\mu_t^i$ into a compact task representation $z_t^i$, and the generalized decoders $(R_\phi, T_\varphi)$ that predict the reward and next state conditioned on $z_t^i$. Then, the inferred task representation is injected as a contextual condition to the causal transformer to guide task-oriented sequence generation. Finally, we design a self-guided prompt from history trajectories generated by the meta-policy at test time. We select the trajectory segment that yields the largest prediction error on the pretrained world model, aiming to encode task-relevant information complementary to the world model maximally.

## 4 Method

In this section, we present Meta Decision Transformer (Meta-DT), a novel offline meta-RL framework that draws upon advances in conditional sequence modeling with robust task representation learning and a complementary prompt design for efficient generalization across unseen tasks. Fig. 1 illustrates the method overview, and the following subsections present the detailed implementations.

### 4.1 Context-Aware World Model

The key to efficient generalization is how to accurately encode task-relevant information for extrapolating knowledge across tasks, which proves more challenging in RL. A supervised or unsupervised learner is *passively* presented with a fixed dataset and aims to recognize the pattern within that dataset. In contrast, RL typically learns from *active* interactions with the environment, and aims at an optimal policy that receives the maximal return in the environment, rather than only recognizing the pattern within the pre-collected dataset. Since the offline dataset depends on both the task and the behavior policy, the task information could be *entangled* with the features of behavior policies, thus producing biased task inference at test time due to the shift in behavior policies. Taking 2D navigation as an example, tasks differ in goals and the behavior policy is going towards the goal for each task. The algorithm can easily distinguish tasks based on state-action distributions rather than reward functions, leading to extrapolating errors when the behavior policy shifts to random exploration during testing.

To address this intrinsic challenge, we ought to *disentangle* task-relevant information from behavior policies. The transition dynamics, i.e., the reward and state transition functions $p(s', r|s, a)$, completely describes the characteristics of the underlying environment. Naturally, this environment dynamics, also called *world model*, keeps invariant to behavior policies or collected datasets, and could be a promising alternative for accurately inferring task beliefs. Therefore, we introduce a context-aware world model to learn robust task representations that can generalize to unseen environments with varying transition dynamics.

In the typical meta-learning setting, the reward and state transition functions that are unique to each MDP are unknown, but also share some common structure across the task distribution $P(M)$. There exists a true variable that represents either a task description or task identity, but we do not have access to this information. Hence, we use a latent representation $z$ to approximate that variable. For a given task $M_i$, the reward and state transition functions can be approximated by a generalized context-aware world model $W$ that is shared across tasks as

$$W_i(r_t, s_{t+1}|s_t, a_t) \approx W(r_t, s_{t+1}|s_t, a_t; z_t^i). \tag{1}$$

Since we do not have access to the true task description or identity, we need to infer task representation $z_t^i$ given the agent's $h$-step experience up to timestep $t$ collected in task $M_i$ as

$$\mu_t^i = (s_{t-h}, a_{t-h}, r_{t-h}, ..., s_{t-1}, a_{t-1}, r_{t-1}, s_t). \tag{2}$$

The intuition is that the true task belief of the underlying MDP can be abstracted from the agent's recent experiences, analogous to recent studies in meta-RL [41, 36].

We separate the reasoning about the world model into two parts: i) encoding the dynamics-specific information into a latent task representation, and ii) decoding the environment dynamics conditioned on that representation. First, we use a simple yet effective context encoder $E_\psi$ to embed recent experiences into a task representation as $z_t^i = E_\psi(\mu_t^i)$. Second, the decoder contains a generalized reward model $R_\phi$ and state transition model $T_\varphi$. The task representation is augmented into the input of both models to predict the instant reward $\hat{r}_{t+1} = R_\phi(s_t, a_t; z_t^i)$ and next state $\hat{s}_{t+1} = T_\varphi(s_t, a_t; z_t^i)$. Under the assumption that tasks with similar contexts will behave similarly [52, 53], the proposed world model can extrapolate the meta-level knowledge across tasks by accurately capturing task-relevant information from training environments. The context encoder is jointly trained by minimizing the reward and state transition prediction error conditioned on the task representation as

$$\mathcal{L}(\psi, \phi, \varphi) = \mathbb{E}_{\mu_t^i \sim \mathcal{D}_i} \left[ \mathbb{E}_{z_t^i \sim E_\psi(\mu_t^i)} \left[ \left( r_{t+1} - R_\phi(s_t, a_t; z_t^i) \right)^2 + \left( s_{t+1} - T_\varphi(s_t, a_t; z_t^i) \right)^2 \right] \right]. \tag{3}$$

## 4.2 Complementary Prompt

Recent works suggest the prompt framework as an effective paradigm for pretraining transformer-based large models on massive datasets and adapting them to new scenarios with few or no labeled data [54]. Prompt-DT [21] adopts that paradigm to RL problems by prepending a task prompt to the DT's input. At test time, the agent is assumed to access a handful of expert demonstrations to construct the prompt and perform few-shot generalization to new tasks. However, in real-world scenarios, it is expensive and even infeasible to acquire such domain knowledge as expert demonstrations in advance for unseen tasks. Especially in RL, agents typically learn from interacting with an initially unknown environment. Recent studies [36, 23] also suggest that the quality of demonstrations must be high enough to act as a well-constructed prompt. Otherwise, the performance of Prompt-DT may degrade since some medium or random data can easily disrupt the abstraction of task-specific information.

Here, we design a self-guided prompt to achieve more realistic generalization at test time. In the ideal case where the testing policy is optimal, past experiences during evaluation can act as high-quality demonstrations to construct the prompt. Motivated by this, we leverage history trajectories generated by the meta-policy during evaluation as a self-guided prompt to enjoy the power of architecture inductive bias, while eliminating the dependency on expensive domain knowledge. Though, the meta-policy may generate medium or even inferior data during initial training. As meta-learning proceeds, the policy can exhibit increasing generalization capacity via the context-aware world model, thus generating trajectories that gradually approach expert demonstrations.

Both the world model (algorithmic perspective) and the self-guided prompt (architecture perspective) aim to extract task-specific information to guide policy generation across tasks. To facilitate collaboration between these two parts, we force the prompt to maximally complement the world model towards the same goal. We feed all available segments selected from history trajectories to the pretrained world model, and use the segment with the largest prediction error to construct the prompt. In this way, the *complementary* prompt attempts to encode the portion of task-relevant information that the world model struggles to capture effectively.

Formally, the complementary prompt is a sequence segment containing multiple tuples, $(\hat{R}^*, s^*, a^*)$, which are sampled from the agent's history trajectories. Each element with superscript $*$ is associated with the prompt. For a given task $M_i$, we obtain a $k$-step prompt $\tau_i^*$ as

$$\tau_i^* = \left( \hat{R}_j^*, s_j^*, a_j^*, \hat{R}_{j+1}^*, s_{j+1}^*, a_{j+1}^*, ..., \hat{R}_{j+k}^*, s_{j+k}^*, a_{j+k}^* \right),$$

$$\text{where} \quad j = \max_j \sum_{t=j}^{j+k} \left[ \left( r_{t+1} - R_\phi(s_t, a_t; z_t^i) \right)^2 + \left( s_{t+1} - T_\varphi(s_t, a_t; z_t^i) \right)^2 \right]. \tag{4}$$

Following Prompt-DT, we choose a much smaller value of $k$ than the task horizon, ensuring that the prompt only contains the information needed to help identify the task but insufficient information for the agent to imitate. Compared to the world model that *explicitly* learns the reward and transition functions, the complementary prompt can store partial information to *implicitly* capture task dynamics.

### 4.3 Meta-DT Architecture

Our Meta-DT architecture is built on decision transformer [8] and solves the offline meta-RL problem through the lens of conditional sequence modeling with robust task representation learning. We first pretrain the context-aware world model and keep it fixed to produce compact task representations for the downstream meta-training process. For each task $M_i$, we use the context encoder from the pretrained world model to infer the contextual information for each timestep as $z_t^i = E_\psi(\mu_t^i)$. Then, the input of Meta-DT consists of two parts: i) the $k$-step complementary prompt $\tau_i^*$ derived from (4), and ii) the most recent $K$-step history $\tau_i^+$ augmented by the learned task representations as

$$\tau_i^+ = (z_{t-K+1}^i, \hat{R}_{t-K+1}, s_{t-K+1}, a_{t-K+1}, ..., z_t^i, \hat{R}_t, s_t, a_t) \tag{5}$$

The input sequence $(\tau_i^*, \tau_i^+)$ corresponds to $3k + 4K$ tokens in the transformer, and Meta-DT autoregressively outputs $k + K$ actions at heads corresponding to state tokens in the input sequence. During training, we construct the prompt from the top few trajectories that obtain the highest returns in the offline dataset. Meta-DT minimizes errors between the predicted and real actions in the data for the $K$-step history. At test time with a *few-shot* setting, the meta-policy is allowed to interact with the environment for a few episodes, and Meta-DT utilizes these history interactions to construct the self-guided prompt. For the *zero-shot* setting, we ablate the prompt component and directly evaluate Meta-DT on test tasks. Corresponding algorithm pseudocodes are given in Appendix A.

## 5 Experiments

We comprehensively evaluate the generalization capacity of Meta-DT on popular benchmarking domains across various dataset types. In general, we aim to answer the following questions:

- Can Meta-DT achieve consistent performance gain on the few and zero-shot generalization to unseen tasks compared with other strong baselines? (Secs. 5.1 and 5.2)

- How do the context-aware world model, the self-guided prompt design, and the complementary prompt construction affect the generalization performance, respectively? (Sec. 5.3)

- Is Meta-DT robust to the quality of offline datasets? (Sec. 5.4)

**Environments.** We evaluate all tested methods on three classical benchmarks in meta-RL: i) the 2D navigation environment `Point-Robot` [25]; ii) the multi-task MuJoCo control [55, 36], containing `Cheetah-Vel`, `Cheetah-Dir`, `Ant-Dir`, `Hopper-Param`, and `Walker-Param`; and iii) the Meta-World manipulation platform [56], including `Reach`, `Sweep`, and `Door-Lock`. For each environment, we randomly sample a distribution of tasks and divide them into the training set $\mathcal{T}^{\text{train}}$ and test set $\mathcal{T}^{\text{test}}$. On each training task, we use SAC [57] to train a single-task policy independently for collecting datasets. We consider three ways to construct offline datasets: `Medium`, `Mixed`, and `Expert`. More details about environments and datasets are given in Appendix B.

**Baselines.** We compare Meta-DT to four competitive baselines that cover two distinctive paradigms in offline meta-RL: the DT-based 1) `Prompt-DT` [21], 2) `Generalized DT` [22], and the temporal difference-based 3) `CORRO` [24], 4) `CSRO` [25]. More details about baselines are given in Appendix C.

All evaluated methods are carried out with 5 different random seeds, and the mean of the received return is plotted with 95% bootstrapped confidence intervals of the mean (shaded). The standard errors are presented for numerical results. Appendix D gives implementation details of Meta-DT. Appendix E presents hyperparameter analysis on the context horizon $h$, the prompt length $k$, and the number of training tasks. Appendix F shows experimental results on Meta-World domains.

### 5.1 Few-shot Generalization Performance

Note that these baselines work under the few-shot setting, since they require expert demonstrations as task prompts or some warm-start data to infer the task representation. We first compare Meta-DT to them under an aligned few-shot setting, where each method can leverage a fixed number of trajectories for task inference. Fig. 2 and Table 1 present the testing curves and converged performance of Meta-DT and baselines on various domains using Medium datasets under an aligned few-shot setting. In these environments with varying reward functions or state transition dynamics, Meta-DT consistently obtains superior performance regarding data efficiency and final asymptotic

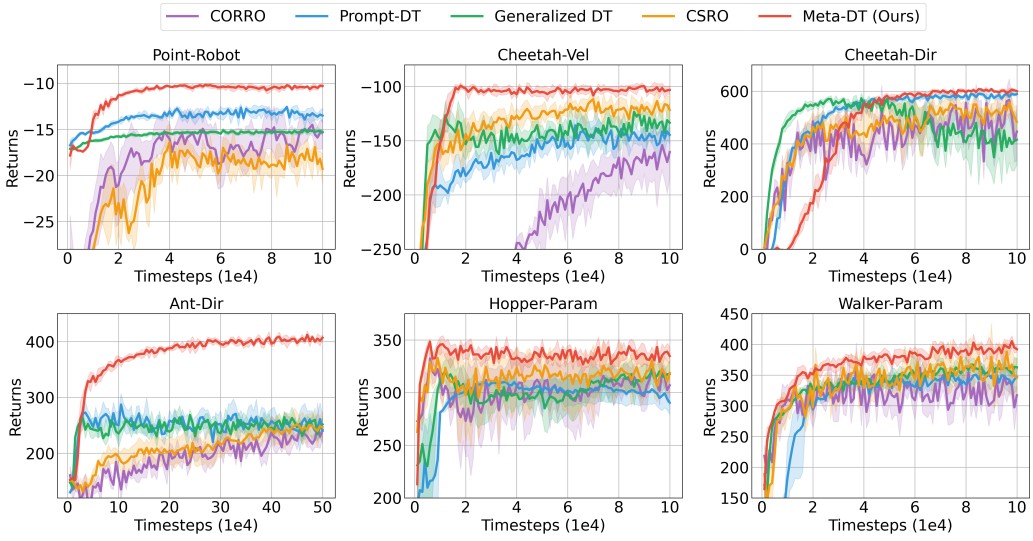

Figure 2: The received return curves averaged over test tasks of Meta-DT and baselines using Medium datasets under an aligned few-shot setting.

Table 1: Few-shot test returns of Meta-DT against baselines using Medium datasets.

| Environment | Prompt-DT | Generalized DT | CORRO | CSRO | Meta-DT |
|---|---|---|---|---|---|
| Point-Robot | -12.58±0.27 | -14.97±0.43 | -14.51±1.65 | -16.39±2.95 | **-10.18**±0.18 |
| Cheetah-Vel | -135.89±19.91 | -123.47±6.88 | -154.12±28.83 | -111.83±9.16 | **-99.28**±3.96 |
| Cheetah-Dir | 590.26±7.28 | 572.94±26.02 | 566.08±96.16 | 556.97±54.59 | **608.17**±4.18 |
| Ant-Dir | 287.23±29.51 | 268.53±12.20 | 246.75±49.33 | 251.66±30.01 | **412.00**±11.53 |
| Hopper-Param | 309.90±5.74 | 320.77±12.82 | 332.37±12.90 | 332.84±11.23 | **348.20**±3.21 |
| Walker-Param | 357.30±18.34 | 368.07±13.79 | 368.02±41.77 | 388.08±24.92 | **405.12**±11.11 |

results. A noteworthy point is that Meta-DT outperforms baselines by a larger margin in relatively complex environments like Ant-Dir, which demonstrates the high generalization capacity of Meta-DT when tackling challenging problems. Moreover, Meta-DT generally exhibits a lower variance during learning, signifying not only superior learning efficiency but also enhanced training stability.

## 5.2 Zero-shot Generalization Performance

Since Meta-DT can derive real-time task representations $z_t$ from its $h$-step experience $\mu_t$ via the pre-trained context encoder, it can achieve zero-shot policy adaptation to unseen tasks. To demonstrate its zero-shot generalization ability, we ablate the prompt component and directly evaluate Meta-DT on test tasks. For a fair comparison, we modify the baselines to an aligned zero-shot setting, where prompt or warm-start data is inaccessible before policy evaluation. All methods can only use samples generated by the trained meta-policy during evaluation to construct task prompts (Prompt-DT), calculate hindsight statistics (Generalized DT), or infer task representations (CORRO and CSRO).

Fig. 3 and Table 2 present the testing curves and converged performance of Meta-DT and baselines on various domains using Medium datasets under an aligned zero-shot setting. Unsurprisingly, most baselines exhibit a collapsed performance (about 20%-35% drop on average) compared to their few-shot counterparts. The main reason is that they usually require expert demonstrations as task prompts or some high-quality warmup data to accurately capture task information. In contrast, Meta-DT can abstract compact task presentations via the context-aware world model, and its performance in zero-shot scenarios drops merely a little (about 5% on average) compared to the few-shot counterpart. The superiority of Meta-DT over all baselines is more pronounced when deployed to zero-shot adaptation scenarios. In addition to its enhanced generalization capacity, this result also demonstrates the superior practicability of Meta-DT with fewer prerequisites in real-world applications.

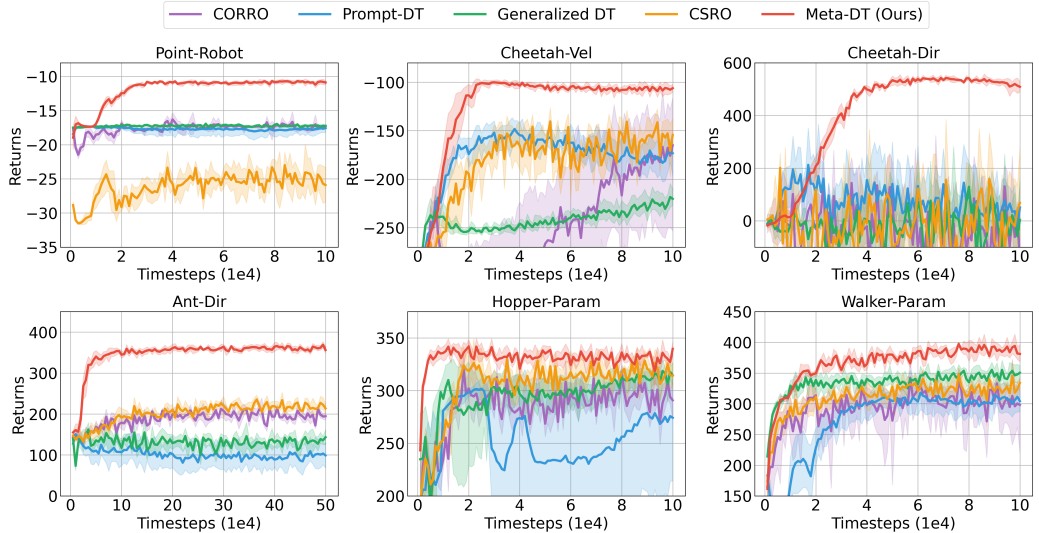

Figure 3: The received return curves averaged over test tasks of Meta-DT and baselines using Medium datasets under an aligned zero-shot setting.

Table 2: Zero-shot test returns of Meta-DT against baselines using Medium datasets. The ↓ denotes the performance drop compared to the few-shot setting.

| Environment | Prompt-DT | Generalized DT | CORRO | CSRO | Meta-DT |
|---|---|---|---|---|---|
| Point-Robot | $-17.3 \pm 0.2$ (↓38%) | $-16.9 \pm 0.1$ (↓13%) | $-16.3 \pm 1.3$ (↓12%) | $-23.0 \pm 2.5$ (↓40%) | $\mathbf{-10.7} \pm 0.3$ (↓5%) |
| Cheetah-Vel | $-148.3 \pm 12.7$ (↓9%) | $-218.4 \pm 15.8$ (↓77%) | $-165.1 \pm 97.4$ (↓7%) | $-140.4 \pm 18.3$ (↓26%) | $\mathbf{-100.1} \pm 2.8$ (↓1%) |
| Cheetah-Dir | $212.4 \pm 185.2$ (↓64%) | $129.2 \pm 106.5$ (↓77%) | $146.4 \pm 169.0$ (↓74%) | $200.9 \pm 97.0$ (↓64%) | $\mathbf{542.4} \pm 13.8$ (↓11%) |
| Ant-Dir | $150.6 \pm 5.4$ (↓48%) | $162.2 \pm 22.4$ (↓40%) | $214.6 \pm 11.0$ (↓13%) | $235.5 \pm 13.3$ (↓6%) | $\mathbf{368.9} \pm 10.6$ (↓10%) |
| Hopper-Param | $301.7 \pm 12.7$ (↓3%) | $318.7 \pm 13.4$ (↓1%) | $320.0 \pm 20.0$ (↓4%) | $329.9 \pm 6.9$ (↓1%) | $\mathbf{342.0} \pm 7.4$ (↓2%) |
| Walker-Param | $317.9 \pm 32.3$ (↓11%) | $355.2 \pm 14.0$ (↓4%) | $342.4 \pm 31.1$ (↓7%) | $347.9 \pm 12.1$ (↓10%) | $\mathbf{397.4} \pm 3.8$ (↓2%) |

## 5.3 Ablation Study

To test the respective contributions of each component, we compare Meta-DT to four ablations: i) w/o_context, it only removes the task representation $z_t^i$ from the input of the casual transformer; ii) w/o_com, it randomly chooses a segment from a candidate trajectory to construct the prompt, i.e., removing the complementary way for constructing the prompt; iii) w/o_prompt, it directly removes the prompt component; iv) DT, it removes all components and degrades to the original DT. For ablations, all other structural modules are kept consistent strictly with the full Meta-DT.

Fig. 4 and Table 3 show ablation results on representative domains using Medium datasets. First, Meta-DT obtains a decreased performance when any component is removed, which demonstrates that all components are essential for Meta-DT's capability and they complement each other. Second, ablating the context incurs a more significant performance drop than ablating the complementary way or the whole prompt. It indicates that task representation learning via the world model plays a more vital role in capturing task-relevant information. Third, the performance order of w/o_prompt < w/o_com < Meta-DT shows that using the self-guided prompt can improve the performance, and leveraging the complementary way to construct the prompt can further achieve a performance gain. Another interesting point is that the performance gain achieved by the self-guided prompt and the complementary way is more significant in complex environments like Ant-Dir than in simpler ones like Point-Robot. It again verifies our superiority in tackling challenging tasks.

## 5.4 Robustness to the Quality of Offline Datasets

A good offline algorithm ought to be robust to different types of datasets that involve a wide range of behavioral policies [58]. To test how Meta-DT performs with data of different qualities, we also conduct experiments on the Mixed and Expert datasets. Figs. 5-6 present test return curves of

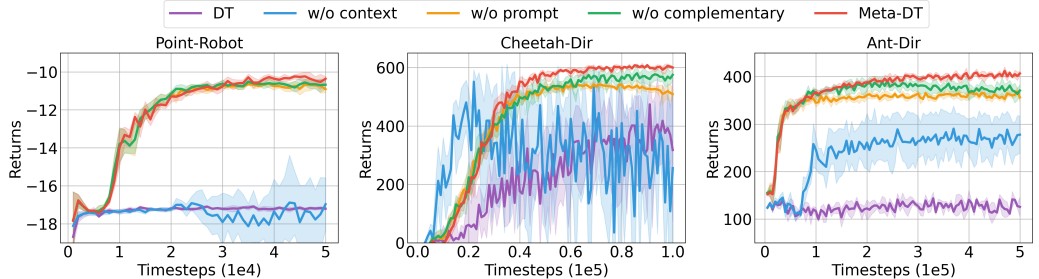

Figure 4: Test return curves of Meta-DT ablations using Medium datasets. w/o_context removes task representation, w/o_com removes the complementary way, and w/o_prompt removes the prompt.

Table 3: Test returns of Meta-DT ablations using Medium datasets.

| Environment | w/o_context | w/o_com | w/o_prompt | DT | Meta-DT |
|---|---|---|---|---|---|
| Point-Robot | -16.44±1.93 | -10.32±0.15 | -10.61±0.15 | -17.04±0.17 | **-10.18**±0.18 |
| Cheetah-Dir | 551.30±55.04 | 580.34±30.84 | 542.44±13.84 | 473.75±104.09 | **608.18**±4.18 |
| Ant-Dir | 290.44±54.36 | 390.70±11.23 | 368.94±10.56 | 143.64±15.36 | **412.00**±11.53 |

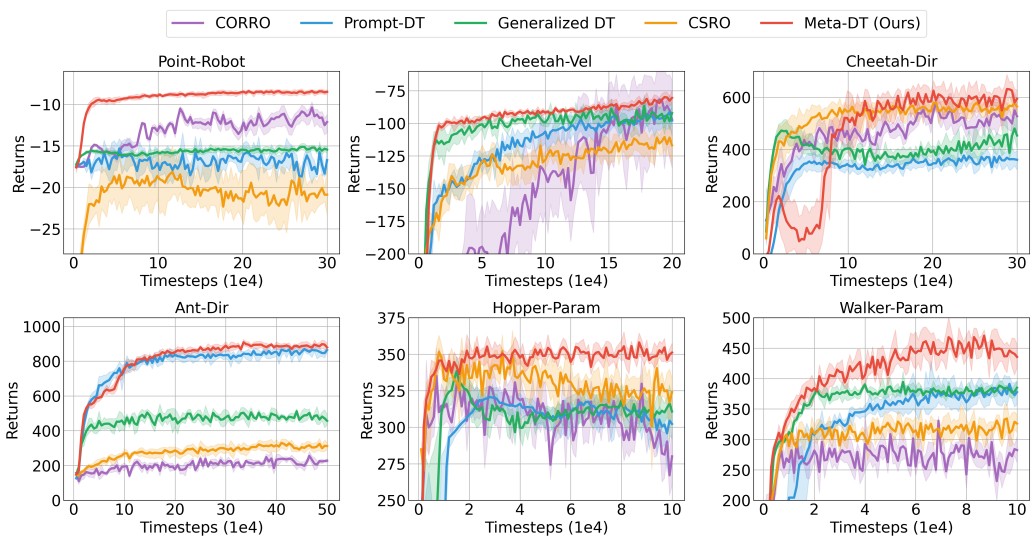

Figure 5: Few-shot test curves of Meta-DT and baselines using Mixed datasets.

Meta-DT and baselines with a few-shot setting, and Table 4 shows corresponding converged results. In Mixed datasets, Meta-DT still outperforms all baselines by a large margin, especially in complex environments like Ant-Dir and Walker-Param.

In Expert datasets, Meta-DT exhibits superior generalization capacity than the three baselines of Generalized DT, CORRO, and CSRO. Compared to Prompt-DT, Meta-DT obtains significantly better performance in Point-Robot, Cheetah-Vel, and Ant-Dir, and obtains comparable performance in the other three environments. This phenomenon is because Prompt-DT is sensitive to the quality of prompt demonstrations. The performance can drop a lot when provided with prompts from Medium or Mixed datasets, also mentioned in the original paper [21] and subsequent studies [36]. Hence, Prompt-DT can achieve satisfactory performance only when expert demonstrations are available at test time. When tackling offline tasks with lower data qualities, Prompt-DT would induce a significant performance gap compared to Meta-DT. In summary, the above results verify that our method is robust to the dataset quality and is more practical with fewer prerequisites in real-world scenarios.

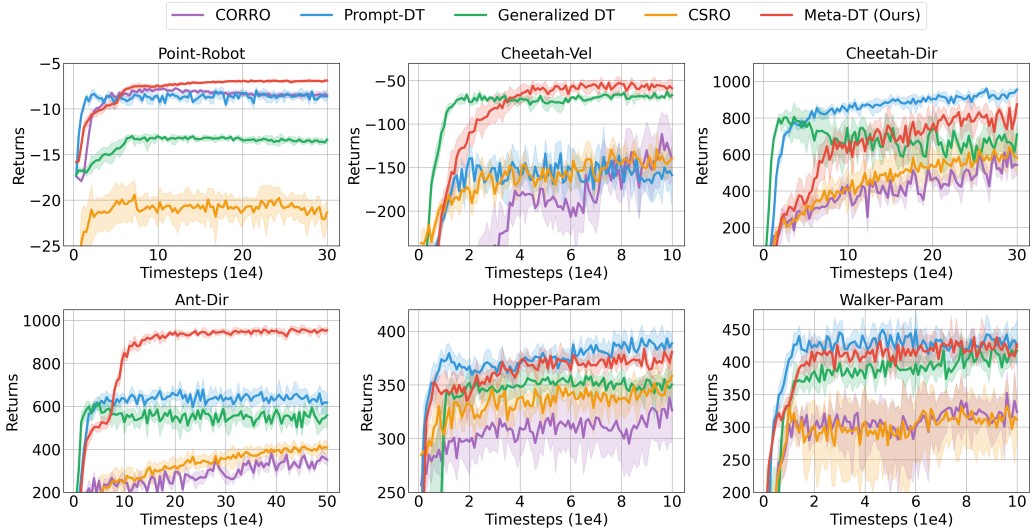

Figure 6: Few-shot test curves of Meta-DT and baselines using Expert datasets.

Table 4: Few-shot test returns of Meta-DT against baselines using Mixed and Expert datasets.

| Mixed | Prompt-DT | Generalized DT | CORRO | CSRO | Meta-DT |
|---|---|---|---|---|---|
| Point-Robot | -15.31±1.52 | -15.10±0.52 | -10.38±1.29 | -18.14±3.75 | **-8.39**±0.28 |
| Cheetah-Vel | -91.34±14.87 | -86.52±10.62 | -81.59±37.17 | -110.55±12.01 | **-79.90**±5.69 |
| Cheetah-Dir | 376.66±33.00 | 480.49±48.38 | 577.80±24.87 | 579.90±15.55 | **631.34**±83.96 |
| Ant-Dir | 869.58±22.64 | 511.97±23.86 | 255.49±32.64 | 330.15±33.77 | **908.48**±20.73 |
| Hopper-Param | 320.76±10.58 | 338.36±14.07 | 335.12±26.73 | 351.96±13.61 | **358.05**±10.69 |
| Walker-Param | 391.24±20.28 | 394.32±19.78 | 330.54±14.07 | 334.34±28.57 | **470.36**±12.85 |

| Expert | Prompt-DT | Generalized DT | CORRO | CSRO | Meta-DT |
|---|---|---|---|---|---|
| Point-Robot | -7.99±0.46 | -12.99±0.34 | -7.76±0.18 | -19.42±2.10 | **-6.90**±0.11 |
| Cheetah-Vel | -133.78±18.24 | -62.95±3.42 | -111.47±36.97 | -129.00±24.24 | **-52.42**±8.11 |
| Cheetah-Dir | **960.32**±17.07 | 806.30±124.41 | 628.64±61.11 | 641.05±129.54 | 874.91±73.45 |
| Ant-Dir | 678.07±68.74 | 613.59±49.22 | 381.42±13.83 | 417.37±39.70 | **961.27**±18.07 |
| Hopper-Param | **393.79**±11.44 | 358.56±11.75 | 338.17±47.36 | 358.29±16.25 | 383.51±8.99 |
| Walker-Param | **449.15**±37.53 | 421.96±40.70 | 352.02±44.99 | 336.89±16.71 | 437.79±18.21 |

## 6   Conclusions, Limitations, and Future Work

In the paper, we tackle the offline meta-RL challenge via leveraging advances in sequence modeling with scalable transformers, marking a step toward developing highly capable generalists akin to those in the language and vision communities. Improvements in few and zero-shot generalization capacities highlight the potential impact of our robust task representation learning and self-guided complementary prompt design. Without requiring any expert demonstration or domain knowledge at test time, our method exhibits superior practicability with fewer prerequisites in real-world scenarios.

Though, our method requires a two-phase process of pre-training the world model and training the causal transformer. A promising improvement is to develop a unified framework that simultaneously abstracts the task representation and learns the meta-policy, akin to in-context learning [59]. Also, our generalist model is trained on relatively lightweight datasets compared to popular large models. A crucial future step is to deploy our model on significantly larger datasets, unlocking the scaling law with the transformer architecture. This also aligns with the urgent trend that RL practitioners are striving to break through. Another interesting line is to leverage self-supervised learning [15, 60] to facilitate task representation learning at scale. We leave these as future work.

## Acknowledgements

We thank Zican Hu and Zhenhong Sun for their helpful discussions. This work was supported by the National Natural Science Foundation of China (Nos. 62376122 & 62073160), the Nanjing University Integrated Research Platform of the Ministry of Education-Top Talents Program, the Beijing Natural Science Foundation under Grant 4232056, the Beijing Nova Program under Grant 20240484514, and the Youth Innovation Promotion Association CAS under Grant 2021132.

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

# Appendix A. Algorithm Pesudocodes

Based on the implementations in Sec. 4, this section gives the brief procedures of our method. First, Algorithm 1 presents the pretraining of the context-aware world model. Then, Algorithm 2 shows the pipeline of training Meta-DT, where the sub-procedure of generating the complementary prompt is given in Algorithm 3. Finally, Algorithm 4 and Algorithm 5 show the few-shot and zero-shot evaluations on test tasks, respectively.

---

**Algorithm 1:** Pretraining the Context-Aware World Model

---

**Input:** Training tasks $\mathcal{T}^{\text{train}}$ and corresponding offline datasets $\mathcal{D}^{\text{train}}$
         Context encoder $E_\psi$;   Reward decoder $R_\phi$;   Transition decoder $T_\varphi$
         Context horizon $h$;     Batch size $B$

1 **for** *each iteration* **do**
2    **for** $b = 1, ..., B$ **do**
3      Sample a task $M_i \sim \mathcal{T}^{\text{train}}$ and obtain the corresponding dataset $\mathcal{D}_i$ from $\mathcal{D}^{\text{train}}$
4      Sample a whole trajectory $(s_0^i, a_0^i, r_0^i, s_1^i, a_1^i, r_1^i, ...)$ from $\mathcal{D}_i$
5      Sample a transition tuple $(s_t, a_t, r_t, s_{t+1})$ with randomly selected $t$
6      Obtain its $h$-step history $\mu_t^i = (s_{t-h}, a_{t-h}, r_{t-h}, ..., s_{t-1}, a_{t-1}, r_{t-1}, s_t)$
7      Compute the context $z_t^i = E_\psi(\mu_t^i)$
8      Compute the predicted reward $\hat{r}_t = R_\phi(s_t, a_t; z_t^i)$ and next state $\hat{s}_{t+1} = T_\varphi(s_t, a_t; z_t^i)$
9    **end**
10    Update $E_\psi$, $R_\phi$, and $T_\varphi$ using the loss as
      $\mathcal{L}(\psi, \phi, \varphi) = \frac{1}{B} \sum \left[ \left( r_{t+1} - R_\phi(s_t, a_t; z_t^i) \right)^2 + \left( s_{t+1} - T_\varphi(s_t, a_t; z_t^i) \right)^2 \right]$
11 **end**

---

**Algorithm 2:** Training Meta Decision Transformer

---

**Input:** Training tasks $\mathcal{T}^{\text{train}}$ and corresponding offline datasets $\mathcal{D}^{\text{train}}$
         Causal transformer $F_\theta$;   Pretrained context encoder $E_\psi$ and decoders $R_\phi, T_\varphi$
         Context horizon $h$;     Trajectory length $K$;     Batch size $B$

1 **for** *each task* $M_i \in \mathcal{T}^{train}$ **do**
2    Construct the demo dataset $\mathcal{D}_i^{\text{demo}}$ using the top few trajectories with the highest returns in $\mathcal{D}_i$
3 **end**
4 **for** *each iteration* **do**
5    **for** $b = 1, 2, ..., B$ **do**
6      Sample a task $M_i \sim \mathcal{T}^{\text{train}}$ and obtain the corresponding dataset $\mathcal{D}_i$ from $\mathcal{D}^{\text{train}}$
7      Sample a trajectory $\tau_i$ of length $K$ from $\mathcal{D}_i$, $\tau_i = \{(\hat{R}, s, a)\}_{t-K+1}^t$
8      Infer the context of each timestep $t$ from its $h$-step history using the pretrained context
        encoder as $z_t = E_\psi(s_{t-h}, a_{t-h}, r_{t-h}, ..., s_{t-1}, a_{t-1}, r_{t-1}, s_t)$
9      Augment the trajectory $\tau_i$ with per-step context as $\tau_i^+ = \{(z, \hat{R}, s, a)\}_{t-K+1}^t$
10      Sample a prompt $\tau_i^* = GetPrompt\left( \mathcal{D}_i^{\text{demo}}, E_\psi, R_\phi, T_\varphi \right)$ as in Algorithm 3
11      Get input $\tau_i^{\text{input}} = (\tau_i^*, \tau_i^+)$
12    **end**
13    Get a batch $\mathcal{B} = \{\tau_i^{\text{input}}\}_{b=1}^B$
14    $a^{\text{pred}} = F_\theta(\tau_i^{\text{input}}), \forall \tau_i^{\text{input}} \in \mathcal{B}$
15    $\mathcal{L}(\theta) = \frac{1}{B} \sum_{\tau_i^{\text{input}} \in \mathcal{B}} (a - a^{\text{pred}})^2$
16    Few or Zero-Shot Evaluation along with training
17 **end**

---

---

**Algorithm 3:** Complementary Prompt Generation (*GetPrompt*)

---

**Input:** Demo Dataset $\mathcal{D}_i^{\text{demo}}$;     Prompt length $k$
       Pretrained context encoder $E_\psi$ and decoders $R_\phi, T_\varphi$

1  Sample a trajectory $(s_0, a_0, r_0, ..., s_t, a_t, r_t, ...)$ from $\mathcal{D}_i^{\text{demo}}$
2  Get the segment with the largest prediction error on the pretrained world model as

$$j = \max_j \sum_{t=j}^{j+k} \left[ \left(r_{t+1} - R_\phi(s_t, a_t; z_t^i)\right)^2 + \left(s_{t+1} - T_\varphi(s_t, a_t; z_t^i)\right)^2 \right], \text{ where } z_t^i = E_\psi(\tau_t^i)$$

3  Obtain the $k$-step prompt $\tau_i^* = \left( \hat{R}_j^*, s_j^*, a_j^*, \hat{R}_{j+1}^*, s_{j+1}^*, a_{j+1}^*, ..., \hat{R}_{j+k}^*, s_{j+k}^*, a_{j+k}^* \right)$

---

---

**Algorithm 4:** Meta-DT Few-Shot Evaluation on Test Tasks

---

**Input:** Test tasks $\mathcal{T}^{\text{test}}$;     Target return $G^*$;     Evaluation episodes $N$
       Context horizon $h$;    Trajectory length $K$;    Prompt length $k$
       Trained meta-DT $F_\theta$;    Pretrained context encoder $E_\psi$ and decoders $R_\phi, T_\varphi$

1  **for** *each task $M_i \in \mathcal{T}^{test}$* **do**
2   Initialize the demo dataset $\mathcal{D}_i^{\text{demo}} = \varnothing$
3   **for** $n = 1, ..., N$ **do**
4    Initialize desired return $\hat{R} = G_i^*$
5    **for** *each timestep $t$* **do**
6     Infer the context from $h$-step history using $E_\psi$ as
          $z_t = E_\psi(s_{t-H+1}, a_{t-H+1}, r_{t-H+1}, ..., s_{t-1}, a_{t-1}, r_{t-1}, s_t)$
7     Augment the trajectory $\tau_i$ with context as $\tau_i^+ = \{(z, \hat{R}, s, a)\}_{t-K+1}^t$
8     Sample a prompt $\tau_i^* = GetPrompt\left(\mathcal{D}_i^{\text{demo}}, E_\psi, R_\phi, T_\varphi\right)$ as in Algorithm 3
9     Get action $a = F_\theta((\tau_i^+, \tau_i^*))$
10    Step env. and get feedback $s, a, r, \hat{R} \leftarrow \hat{R} - r$
11    Append $(\hat{R}, s, a)$ to recent history $\tau_i$
12    **end**
13   Append the whole trajectory $\tau_i$ to the demo dataset as $\mathcal{D}_i^{\text{demo}} \leftarrow \mathcal{D}_i^{\text{demo}} \cup \tau_i$
14   **end**
15 **end**

---

---

**Algorithm 5:** Meta-DT Zero-Shot Evaluation on Test Tasks

---

**Input:** Test tasks $\mathcal{T}^{\text{test}}$;     Target return $G^*$
       Context horizon $h$;    Trajectory length $K$
       Trained meta-DT $F_\theta$;    Pretrained context encoder $E_\psi$

1  **for** *each task $M_i \in \mathcal{T}^{test}$* **do**
2   Initialize desired return $\hat{R} = G_i^*$
3   **for** *each timestep $t$* **do**
4    Infer the context from $h$-step history using $E_\psi$ as
         $z_t = E_\psi(s_{t-H+1}, a_{t-H+1}, r_{t-H+1}, ..., s_{t-1}, a_{t-1}, r_{t-1}, s_t)$
5    Augment the trajectory $\tau_i$ with context as $\tau_i^+ = \{(z, \hat{R}, s, a)\}_{t-K+1}^t$
6    Get action $a = F_\theta(\tau_i^+)$
7    Step env. and get feedback $s, a, r, \hat{R} \leftarrow \hat{R} - r$
8    Append $(\hat{R}, s, a)$ to recent history $\tau_i$
9   **end**
10 **end**

---

## Appendix B. The Details of Environments and Dataset Construction

In this section, we show details of evaluation environments over a variety of testbeds, as well as the offline dataset collection process conducted on these environments.

### B.1. The Details of Environments

Following the mainstream studies in offline meta-RL [39, 24], we adopt three classical benchmarks: the 2D navigation [25], the multi-task MuJoCo control [55, 36], and the Meta-World [56]. We evaluate all tested methods on the following environments as

- **Point-Robot**: a problem of a point agent navigating to a given goal position in the 2D space. The observation is the 2D coordinate of the robot and the goal is not included in the observation. The action space is $[-0.1, 0.1]^2$ with each dimension corresponding to the moving distance in the horizontal and vertical directions. The reward function is defined as the negative distance between the point agent and the goal location. Each learning episode always starts from the fixed origin and terminates at the horizon of 20. Tasks differ in goal positions that are uniformly distributed in a unit square, resulting in the variation of the reward functions.

- **Cheetah-Vel**, **Cheetah-Dir**, **Ant-Dir**: multi-task MuJoCo continuous control benchmarks in which the reward functions differ across tasks. Cheetah-Vel requires a planar cheetah robot to run at a particular velocity in the positive $x$-direction, and the reward function is negatively correlated with the absolute value between the current velocity of the agent and a goal. Cheetah-Dir/Ant-Dir is to control a 2D cheetah/3D quadruped ant robot to move in a specific direction, and the reward function is the cosine product of the agent's velocity and the goal direction. The goal velocity and direction are uniformly sampled from the distribution $U[0.075, 3.0]$ for Cheetah-Vel and $U[0, 2\pi]$ for Ant-Dir. In Cheetah-Dir, the goal directions are limited to forward and backward. The maximal episode step is set to 200.

- **Hopper-Param**, **Walker-Param**: multi-task MuJoCo benchmarks where tasks differ in state transition dynamics. The two benchmarks control a one-legged hopper and a two-legged walker robot to run as fast as possible. The reward function is proportional to the running velocity in the positive $x$-direction, which remains consistent for different tasks. In both domains, the physical parameters of body mass, inertia, damping, and friction are randomized across tasks. The agent needs to move forward with varying environment dynamics to accomplish the task. The maximal episode step is set to 200 for both.

- **Reach**, **Sweep**, **Door-Lock**: three typical environments from the robotic manipulation benchmark Meta-World. Reach, Sweep, and Door-Lock control a robotic arm to reach a goal location in 3D space, to sweep a puck off the table, and to lock the door by rotating the lock clockwise, respectively. Tasks are assigned with random goal object positions for Reach, with random puck positions for Sweep, and with random door positions for Door-Lock. The variations in goal/puck/door positions are not provided in the observation, forcing meta-RL algorithms to adapt to the goal through trial-and-error. The maximal episode step is set to 500.

For the Point-Robot and MuJoCo environments, we sample 45 tasks for training and another 5 held-out tasks for testing. For Meta-World environments, we sample 15 tasks for training and 5 held-out tasks for testing.

### B.2. The Details of Datasets

We employ the soft actor-critic (SAC) [57] algorithm to train a policy independently for each task. Table 5 shows the detailed hyperparameters for training the SAC agents in each environment. Figs. 7-8 show the learning curves of independently training the SAC agents for each training and testing task in all evaluation domains. During training, we periodically save the policy checkpoints to generate various types of offline datasets as

- **Medium**: using a medium policy that achieves a one-third to one-second score of an expert policy. All trajectories are generated by this medium policy.

- **Expert**: using an expert policy that obtains the optimal return in the environment. In practice, we load the last policy checkpoint after the training has converged, and use this expert policy to generate all trajectories.

- **Mixed**: a mixed dataset of the Medium and Expert types. We use a combination of the medium (70%) and expert (30%) datasets collected as above to generate the mixed dataset.

Specifically, we generate 100 trajectories for the Point-Robot and MuJoCo environments, and 300 trajectories for the Meta-World environments, using the saved checkpoint policies to construct corresponding offline datasets.

### B.3. Further discussions

The above environments are relatively homogenous in nature. Our generalist model is trained on relatively lightweight datasets compared to popular large models. The urgent trend that RL practitioners are striving to break through is to deploy on significantly large datasets with more diversity, unlocking the scaling law with the transformer architecture. It would be interesting to see how Meta-DT generalizes across worlds and tasks with more diversity, like training on $K$-levels of an Atari game and generalizing to the remaining $N - K$ levels of well-suited Atari games.

The challenges in extending to such paradigms might include: i) how to collect large-scale datasets with sufficient diversity and high quality, ii) how to tackle the heterogenous task diversity in hard cases, and iii) how to scale RL models to very large network architectures like in large language or visual models. To tackle these potential challenges, we conjecture several promising solutions including: i) leveraging self-supervised learning to facilitate task representation learning at scale, ii) enabling efficient prompt design or prompt tuning to facilitate in-context RL, and iii) incorporating effective network architectures like mixture-of-experts to handle heterogenous domain diversity.

Table 5: Hyperparameters of SAC used to collect multi-task datasets.

| Environments | Training steps | Warmup steps | Save frequency | Learning rate | Soft update | Discount factor | Entropy ratio |
|---|---|---|---|---|---|---|---|
| Point-Robot | 2000 | 100 | 40 | 3e-4 | 0.005 | 0.99 | 0.2 |
| Cheetah-Dir | 500000 | 10000 | 10000 | 3e-4 | 0.005 | 0.99 | 0.2 |
| Cheetah-Vel | 500000 | 2000 | 10000 | 3e-4 | 0.005 | 0.99 | 0.2 |
| Ant-Dir | 500000 | 2000 | 10000 | 3e-4 | 0.005 | 0.99 | 0.2 |
| Hopper-Param | 500000 | 10000 | 10000 | 3e-4 | 0.005 | 0.99 | 0.2 |
| Walker-Param | 500000 | 2000 | 10000 | 3e-4 | 0.005 | 0.99 | 0.2 |
| Reach | 200000 | 5000 | 1000 | 3e-4 | 0.005 | 0.99 | 0.2 |
| Sweep | 200000 | 5000 | 1000 | 3e-4 | 0.005 | 0.99 | 0.2 |
| Door-Lock | 200000 | 5000 | 1000 | 3e-4 | 0.005 | 0.99 | 0.2 |

## Appendix C. The Details of Baselines

This section gives the details of the representative five baselines, including three DT-based approaches, one temporal difference-based approach, and one diffusion-based approach. These baselines are thoughtfully selected to encompass the three distinctive categories of offline meta-RL methods. Also, since our method Meta-DT belongs to the DT-based category, we adopt more approaches from this kind as our baselines. The baselines are introduced as follows:

- **Prompt-DT** [21], is a DT-based method that leverages the sequential modeling ability of the Transformer architecture and the prompt framework to achieve few-shot adaptation in offline RL. It designs the trajectory prompt, which contains segments of the few-shot demonstrations, and encodes task-specific information to guide policy generation. At test time, it assumes that the agent can assess a handful of expert demonstrations to construct the prompt.

- **Generalized DT** [22], is a DT-based method that formulates multi-task learning as a hindsight information matching (HIM) problem, i.e., training policies that can output the rest of the trajectory to match some statistics of future information. It defines offline multi-task state-marginal matching and imitation learning as two generic HIM problems to evaluate the proposed Categorical DT and Bi-directional DT.

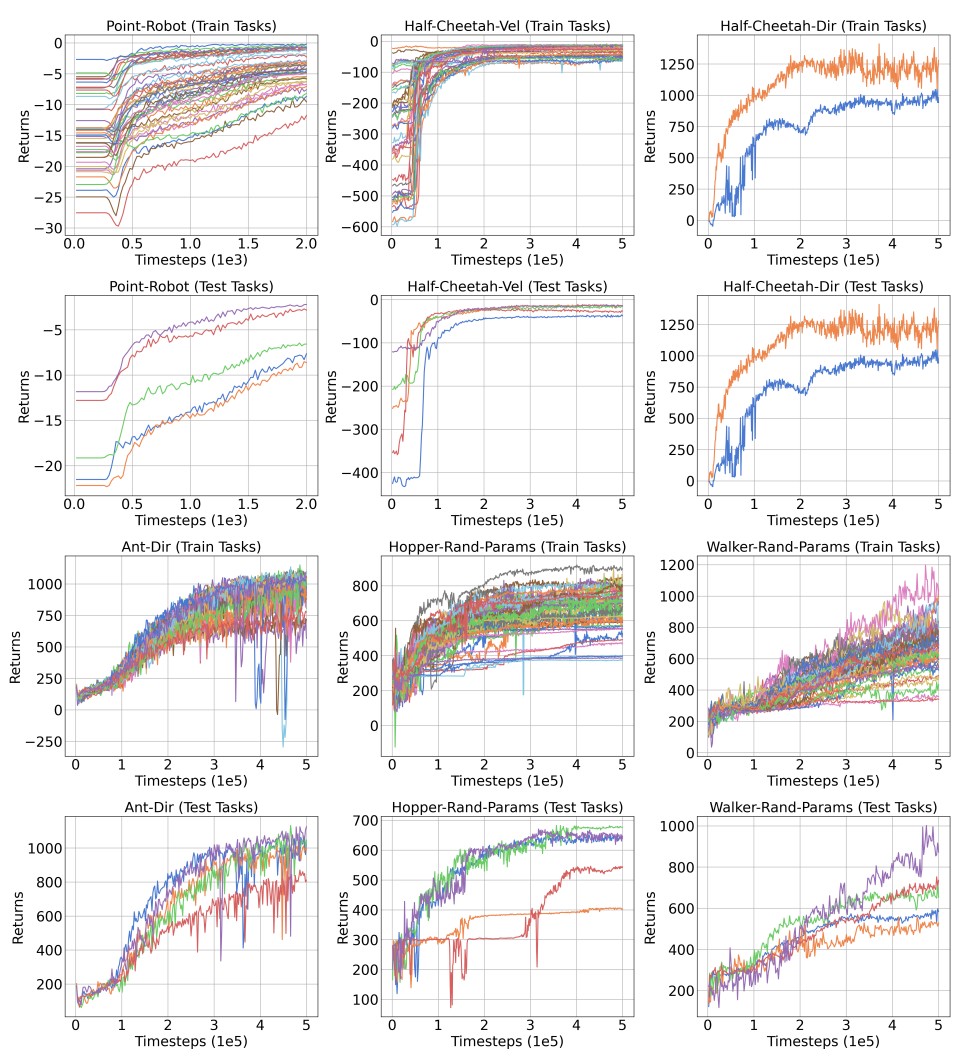

Figure 7: Learning curves of training independent SAC agents in Point-Robot and MuJoCo domains.

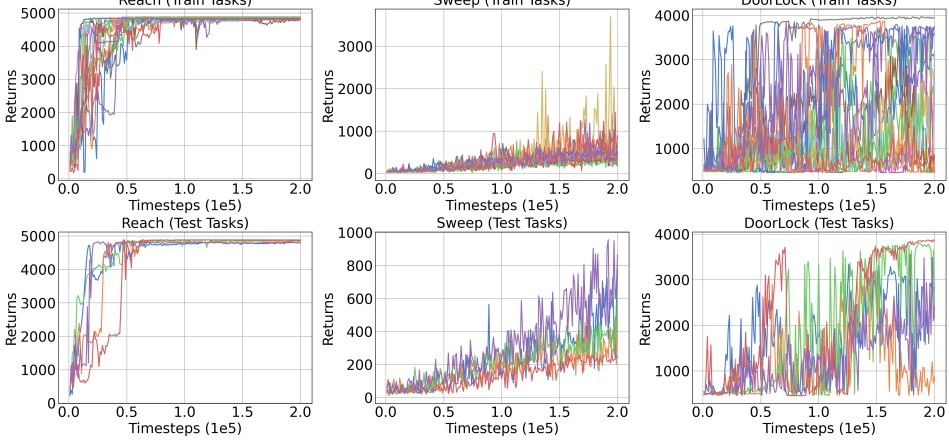

Figure 8: Learning curves of independently training the SAC agents for each task in Meta-World.

- **CORRO** [24], is a temporal difference-based method with a contrastive learning framework to learn task representations that are robust to the distribution mismatch of behavior policies in training and testing. It formalizes the learning objective as mutual information maximization between the representation and task, to maximally eliminate the influence of behavior policies from task representations. At test time, it needs to collect a context trajectory with an arbitrary policy to infer the task representation.

- **CSRO** [25], is a temporal difference-based method that proposes a max-min mutual information representation learning framework to mitigate the distribution discrepancy between the contexts used for training (from the behavior policy) and testing (for the exploration policy). At test time, it explores the new task with a non-prior strategy and collects few-shot context data to infer the task representation. It demonstrates superior performance than prior context-based meta-RL methods.

Note that these baselines can only work under the few-shot setting, since they require several expert demonstrations as the task prompt or some warm-start data to infer the task representation. We first compare Meta-DT to them under an aligned few-shot setting, where each method can leverage a fixed number of trajectories for task inference.

Further, we demonstrate the zero-shot generalization capability of Meta-DT and baselines by directly evaluating the meta-policy on the unseen task without collecting any data in advance. For Meta-DT, we ablate the prompt component and derive real-time task representations from the history trajectory via the pre-trained context encoder. For a fair comparison, we modify the baselines to an aligned zero-shot setting, where prompt demonstrations or warm-start data are inaccessible before policy evaluation. All these baselines can only use samples generated by the trained meta-policy during evaluation to construct task prompts (Prompt-DT), calculate hindsight statistics (Generalized DT), or infer task representations (CORRO and CSRO).

## Appendix D. Implementation Details of Meta-DT

**Network Architecture.** In this paper, we use simple network structures for the context-aware world model, including the context encoder, the reward decoder, and the state transition decoder. Specifically, the context encoder contains a fully-connected multi-layer perceptron (MLP) and a GRU network with ReLU as the activation function. The GRU cell encodes the agent's $h$-step history $(s_{t-h}, a_{t-h}, r_{t-h}, ..., s_{t-1}, a_{t-1}, r_{t-1}, s_t)$ into a 128-dimensional vector, and the MLP transforms this vector into a 16-dimensional embedding, i.e., the task representation $z$. The reward decoder is an MLP that takes as input the $(s, a, s', z)$ tuple and predicts the reward $r$ through two 128-dimensional hidden layers. Similarly, the state transition decoder is an MLP that takes as input the $(s, a, r, z)$ tuple and predicts the next state $s'$ through two 128-dimensional hidden layers.

We implement the proposed Meta-DT based on the official codebase released by DT [8] (https://github.com/kzl/decision-transformer). We follow most of the hyperparameters as they did. Specifically, task representations $z$, returns $\hat{R}$, states $s$, and actions $a$ are fed into modality-specific linear embeddings, and the same positional episodic timestep encoding is added to tokens corresponding to the same timestep. Tokens are fed into a GPT architecture which predicts actions autoregressively using a causal self-attention mask. In summary, Table 6 shows the details of network structures.

**Algorithm Hyperparameters.** We evaluate the proposed Meta-DT algorithm on various testbeds with different types of offline datasets. Each report unit involves running on one environment (Point-Robot, Cheetah-Vel, Cheetah-Dir, Ant-Dir, Hopper-Param, Walker-Param, Reach) with one dataset (Medium, Expert, Mixed). Some common hyperparameters across all report units are set as: optimizer Adam, weight decay 1e-4, linear warmup steps for learning rate decay 10000, gradient norm clip 0.25, dropout 0.1, and batch size 128. Table 7 presents the detailed hyperparameters of Meta-DT trained on the Point-Robot and MuJoCo domains with the Medium, Expert, and Mixed datasets. Table 8 presents the detailed hyperparameters of Meta-DT trained on Meta-World environments with the Medium datasets.

**Compute.** We train our models on one Nvidia RTX4080 GPU with the Intel Core i9-10900X CPU and 256G RAM. Pretraining the context-aware world model and the causal transformer costs about 0.5-6 hours and 1-8 hours, respectively, depending on the complexity of the environment.

Table 6: The network configurations used for Meta-DT.

| World Model | Value | Causal Transformer | Value |
|---|---|---|---|
| GRU hidden dim | 128 | layers num | 3 |
| task representation dim | 16 | attention heads num | 1 |
| decoder hidden dim | 128 | embedding dim | 128 |
| decoder hidden layers num | 2 | activation function | ReLU |
| activation function | ReLU | | |

Table 7: Hyperparameters of Meta-DT on Point-Robot and MuJoCo domains with various datasets.

| **Medium** | Point-Robot | Cheetah-Vel | Cheetah-Dir | Ant-Dir | Hopper-Param | Walker-Param |
|---|---|---|---|---|---|---|
| training steps | 100000 | 100000 | 100000 | 500000 | 100000 | 100000 |
| sequence length $K$ | 8 | 20 | 20 | 20 | 20 | 20 |
| context horizon $h$ | 4 | 4 | 4 | 4 | 4 | 4 |
| learning rate | 1e-4 | 5e-5 | 6e-6 | 5e-5 | 5e-5 | 5e-5 |
| prompt length $k$ | 3 | 5 | 5 | 5 | 5 | 5 |
| **Expert** | Point-Robot | Cheetah-Vel | Cheetah-Dir | Ant-Dir | Hopper-Param | Walker-Param |
| training steps | 300000 | 200000 | 300000 | 500000 | 100000 | 100000 |
| sequence length $K$ | 8 | 20 | 20 | 20 | 20 | 20 |
| context horizon $h$ | 4 | 4 | 4 | 4 | 4 | 4 |
| learning rate | 1e-4 | 5e-5 | 5e-5 | 5e-5 | 5e-5 | 5e-5 |
| prompt length $k$ | 3 | 5 | 5 | 5 | 5 | 5 |
| **Mixed** | Point-Robot | Cheetah-Vel | Cheetah-Dir | Ant-Dir | Hopper-Param | Walker-Param |
| training steps | 300000 | 200000 | 300000 | 500000 | 100000 | 100000 |
| sequence length $K$ | 8 | 20 | 20 | 20 | 20 | 20 |
| context horizon $h$ | 4 | 4 | 4 | 4 | 4 | 4 |
| learning rate | 1e-4 | 5e-5 | 5e-5 | 5e-5 | 5e-5 | 5e-5 |
| prompt length $k$ | 3 | 5 | 5 | 5 | 5 | 5 |

Table 8: Hyperparameters of Meta-DT trained on Meta-World domains with Medium datasets.

| Hyperparameters | Reach | Sweep | Door-Lock |
|---|---|---|---|
| training steps | 500000 | 500000 | 500000 |
| sequence length $K$ | 20 | 20 | 20 |
| context horizon $h$ | 4 | 4 | 4 |
| learning rate | 1e-7 | 1e-6 | 1e-6 |
| prompt length $k$ | 5 | 5 | 5 |

## Appendix E. Hyperparameter Analysis

**Context horizon** $h$. As shown by the ablation studies in Sec. 5.3, task representation learning via the world model disentanglement plays a more vital role in capturing task-relevant information in Meta-DT. The task representation $z_t^i$ is abstracted from the agent's $h$-step experience $\mu_t^i = (s_{t-h}, a_{t-h}, r_{t-h}, ..., s_{t-1}, a_{t-1}, r_{t-1}, s_t)$. The context horizon $h$ is a key hyperparameter in learning effective task representations. Hence, we conduct additional experiments to analyze the influence of context horizon $h$ on Meta-DT's performance.

Fig. 9 and Table 9 present the testing curves and converged performance of Meta-DT with different values of context horizon $h$ on representative environments. Generally, the performance of Meta-DT

is not sensitive to the pre-defined value of the context horizon. In simple environments like Point-Robot, increasing the context horizon can degrade the final performance a little bit. The most likely reason for this phenomenon is that the context-aware world model might be overfitted with a large value of context horizon. In more complex environments like Cheetah-Dir and Ant-Dir, adopting different values of context horizon leads to very close performance. Hence, throughout the paper, we choose the context horizon as 4 for our method to exploit a lightweight network design while not sacrificing performance.

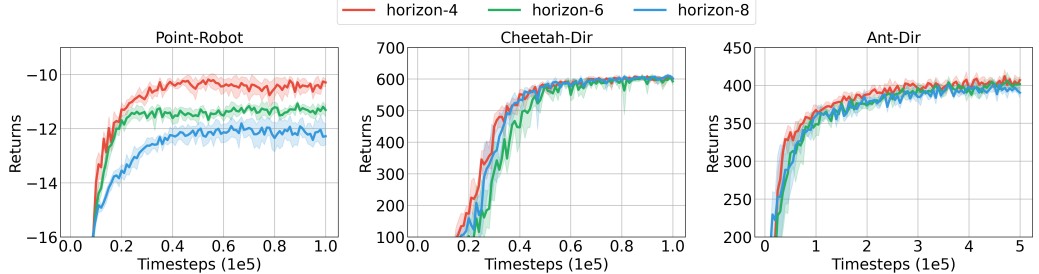

Figure 9: The received return curves averaged over test tasks of Meta-DT with different values of context horizon $h$ using Medium datasets under an aligned few-shot setting.

Table 9: Few-shot test returns averaged over test tasks of Meta-DT with different values of context horizon $h$ using Medium datasets.

| Context Horizon $h$ | 4 | 6 | 8 |
|---|---|---|---|
| Point-Robot | **-10.18**±0.18 | -11.08±0.11 | -11.81±0.31 |
| Cheetah-Dir | 608.18±4.18 | 606.81±11.34 | **610.63**±6.13 |
| Ant-Dir | **412.00**±11.53 | 406.01±12.61 | 401.36±6.65 |

**Prompt length** $k$. Analogously, we also investigate the influence of prompt length $k$ on Meta-DT's performance. Fig. 10 and Table 10 present the testing curves and converged performance of Meta-DT with different values of prompt length $k$ on representative environments. Generally, the performance of Meta-DT is not sensitive to the pre-defined value of prompt length.

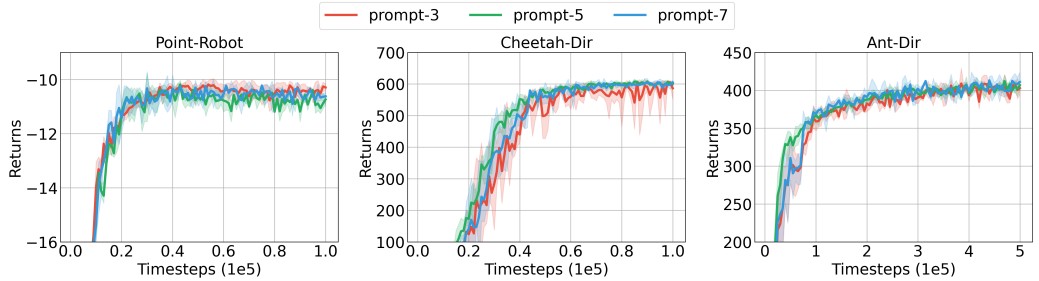

Figure 10: Few-shot return curves of Meta-DT with different prompt length $k$ using Medium datasets.

Table 10: Few-shot test returns of Meta-DT with different prompt length $k$ using Medium datasets.

| prompt length $k$ | 3 | 5 | 7 |
|---|---|---|---|
| Point-Robot | **-10.18**±0.18 | -10.19±0.15 | -10.22±0.04 |
| Cheetah-Dir | 606.69±2.92 | **608.18**±4.18 | 605.09±6.31 |
| Ant-Dir | 408.97±18.01 | 412.00±11.53 | **413.09**±8.82 |

**The number of training tasks**. We evaluate Meta-DT with a different number of training tasks. Fig. 11 and Table 11 present the testing curves and converged performance of Meta-DT with different numbers of training tasks on representative environments using Medium datasets. Generally, increasing the number of training tasks can usually improve the generalization ability to test tasks, especially in harder tasks like Ant-Dir.

The result also matches the intuition of function approximation in machine learning. The "generalization" refers to the question: How can experience with a limited subset of the data space be usefully generalized to produce a good approximation over a much larger subset? In the single-task setting, increasing the valid sample points (before being overfitted) can generally improve the function approximation of the sample space and the generalization to unseen samples at testing. In the meta-learning setting, increasing the valid task points (before being overfitted) can usually boost the function approximation of the task space and the generalization to unseen tasks at testing.

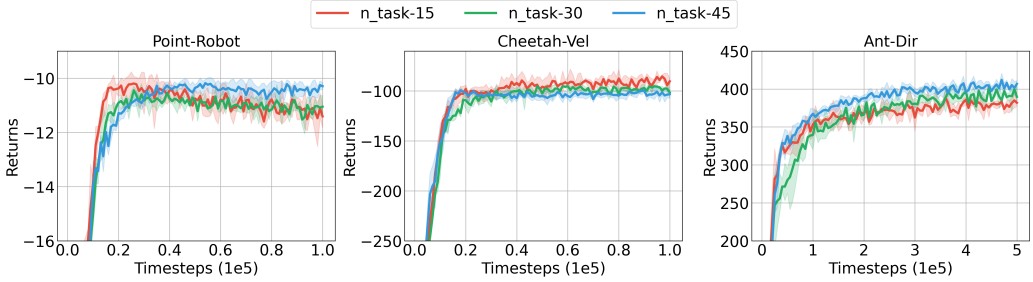

Figure 11: Few-shot return curves of Meta-DT with different numbers of training tasks (Medium).

Table 11: Few-shot test returns of Meta-DT with different numbers of training tasks (Medium).

| Number of training tasks | 15 | 30 | 45 |
|---|---|---|---|
| Point-Robot | -10.20±0.28 | -10.45±0.2 | **-10.18**±0.18 |
| Cheetah-Vel | **-85.42**±1.59 | -95.14±0.68 | -99.28±3.96 |
| Ant-Dir | 388.50±9.02 | 402.71±5.00 | **412.00**±11.53 |

## Appendix F. Experimental Results on Meta-World

Here, we present experimental results on the robotic manipulation benchmark Meta-World [56], including three environments of Reach, Sweep, and Door-Lock. Fig. 12 and Table 12 show the testing curves and converged performance of Meta-DT and baselines using Medium datasets under an aligned few-shot setting. The results show that Meta-DT still performs better than the baselines in these challenging and realistic Meta-World environments.

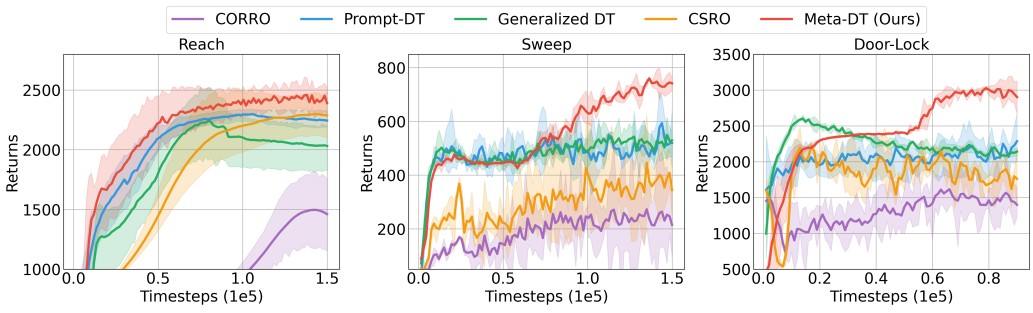

Figure 12: Few-shot return curves of Meta-DT and baselines on Meta-World using Medium datasets.

Table 12: Few-shot test returns of Meta-DT and baselines on Meta-World using Medium datasets.

| Env. | Prompt-DT | Generalized DT | CORRO | CSRO | Meta-DT |
|---|---|---|---|---|---|
| Reach | 2296.55±66.54 | 2241.00±394.37 | 1496.94±402.27 | 2298.69±224.53 | **2458.83**±162.10 |
| Sweep | 593.11±92.10 | 549.17±79.35 | 272.67±77.41 | 446.50±191.46 | **760.18**±4.87 |
| Door-Lock | 2284.95±341.20 | 2601.24±44.42 | 1615.66±248.00 | 2216.67±116.84 | **3025.92**±23.41 |

## Appendix G. Generalization to Out-of-Distribution Tasks

Most meta-RL studies follow the experimental setting as training in a large distribution of tasks and evaluating in a few held-out tasks. The held-out tasks have goals (target position or direction) within the goal range calibrated by training tasks. We desire to test whether Meta-DT enables knowledge extrapolation when handling tasks with goals out of the training range, i.e., the generalization ability in out-of-distribution (OOD) tasks. Hence, we conduct experiments on Ant-Dir to evaluate Meta-DT's generalization ability to OOD tasks. In this case, we sample training tasks with a goal direction of $U[0, 1.5\pi]$ and then test on tasks of $[1.5\pi, 2\pi]$. Fig. 13 and Table 13 show the few-shot return curves and converged performance on OOD test tasks using Medium datasets from Ant-Dir.

Obviously, Meta-DT can still obtain better performance than baselines on OOD test tasks, which again verifies our superiority. Our method extrapolates the meta-level knowledge across tasks by the extrapolation ability of the world model, which is more accurate and robust since the world model is intrinsically invariant to behavior policies or collected datasets. For tasks like Ant-Dir, the world model shares some common structure across the task distribution (even for OOD tasks), e.g., the kinematics principle or locomotion skills. Hence, the extrapolation of the world model also works for OOD test tasks in this case.

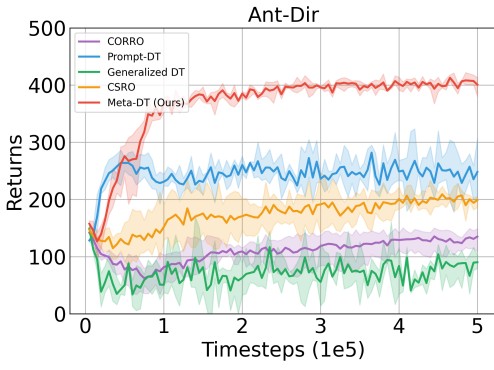

Figure 13: Few-shot return curves of OOD test tasks using Medium datasets from Ant-Dir.

Table 13: Few-shot returns of OOD test tasks using Medium datasets from Ant-Dir.

| Method | OOD Performance |
|---|---|
| Prompt-DT | 281.73±12.29 |
| Generalized DT | 148.83±6.06 |
| CORRO | 141.79±94.20 |
| CSRO | 208.92±15.47 |
| Meta-DT | **412.67**±5.15 |

