# OpenReview forum: "Meta-DT: Offline Meta-RL as Conditional Sequence Modeling with World Model Disentanglement"
_NeurIPS.cc/2024/Conference — NeurIPS 2024 poster_

### Official Review · Reviewer_BWEr · 2024-07-10

**Soundness:** 3
**Presentation:** 3
**Contribution:** 3
**Rating:** 7
**Confidence:** 3

**Summary:**

A new framework is proposed for offline meta-reinforcement learning (meta-RL) that leverages transformers and world model disentanglement to enhance task generalization without the need for expert demonstrations or domain knowledge. The approach, called Meta Decision Transformer (Meta-DT), utilizes a context-aware world model to encode task-relevant information which guides the transformer in generating task-oriented sequences. Experimental results show that Meta-DT achieves superior few-shot and zero-shot generalization across different benchmarks, indicating its practicality and effectiveness in RL applications.

**Strengths:**

Generalization without Experts: Meta-DT effectively generalizes across unseen tasks without requiring expert demonstrations or domain knowledge, which is a significant advancement in the field of meta-reinforcement learning.

World Model Disentanglement: The introduction of a context-aware world model that disentangles task-relevant information from behavior policies enhances the robustness and accuracy of task representation.

Efficient Use of Transformers: The paper creatively applies the transformer architecture to offline meta-RL, leveraging its strong sequential modeling capabilities to improve policy generation and adaptation.

Empirical Validation: The paper includes comprehensive experimental results showing that Meta-DT outperforms existing baselines in few-shot and zero-shot generalization tasks across multiple benchmarks, demonstrating its practical effectiveness.

**Weaknesses:**

1. Are there any environments or types of tasks where Meta-DT's performance is notably limited? What are the challenges in extending the model to such environments, and how might these be addressed in future work?

2. Can you elaborate on the process and benefits of world model disentanglement in Meta-DT? How does this process affect the model's ability to generalize across different tasks, especially in dynamically changing environments?

**Questions:**

Mentioned in the weakness section

**Limitations:**

Yes

---

> ### Author Rebuttal · Authors · 2024-08-06
>
> **Q1. Are there any environments or types of tasks where Meta-DT's performance is notably limited? What are the challenges in extending the model to such environments, and how might these be addressed in future work?**
>
> A1. Thank you for your insightful questions. We evaluated Meta-DT on seven environments adopted from three classical benchmarks in meta-RL. During rebuttal, we also conducted experiments on new evaluation domains, including Humanoid-Dir and two more Meta-World environments (Figure S2 in Global Response). **Results showed the consistent superiority of Meta-DT in these ten environments.**
>
> On the other hand, the above environments are relatively homogenous in nature. As stated in the limitations section, our generalist model is trained on relatively lightweight datasets compared to popular large models. The urgent trend that RL practitioners are striving to break through is to deploy on significantly large datasets with more diversity. As mentioned by Reviewer sbh9, **it would be interesting to see how Meta-DT generalizes across worlds and tasks with more diversity**, like training on $K$-levels of an Atari game and generalizing to the remaining $N-K$ levels of well-suited Atari games.
>
> The **challenges** in extending to such paradigms might include: i) how to collect large-scale datasets with sufficient diversity and high quality, ii) how to tackle the heterogenous task diversity in hard cases, and iii) how to scale RL models to very large network architectures like in large language or visual models. To tackle these potential challenges, we conjecture several **promising solutions** including: i) leveraging self-supervised learning to facilitate task representation learning at scale, ii) enabling efficient prompt design or prompt tuning to facilitate in-context RL, and iii) incorporating effective network architectures like mixture-of-experts to handle heterogenous domain diversity. We will include the above insights into our limitations section, and are looking forward to investigating these promising future directions.
>
>
> **Q2. Elaborate on the process and benefits of world model disentanglement in Meta-DT, and how it affects the model's generalization ability.**
>
> A2. Existing context-based or prompt-based methods usually infer task representations by feeding subsets of experience into a trajectory encoder. However, in RL regimes, since the offline dataset depends on both the task and the behavior policy, **the task information could be entangled with the features of behavior policies**, thus producing biased task inference at test time due to the shift in behavior policies. Taking 2D navigation as an example, tasks differ in goals and the behavior policy is going towards the goal for each task. The algorithm can easily distinguish tasks based on state-action distributions rather than reward functions, leading to extrapolating errors when the behavior policy shifts to random exploration during testing.
>
> Hence, we attempt to find a stable way to accurately disentangle task-relevant information from behavior policies. For RL paradigms, **the world model completely describes the characteristics of the MDP/task, and keeps invariant to behavior policies or collected datasets**. Naturally, it could be a promising alternative for accurately disentangling task beliefs. That is exactly the motivation of our world model disentanglement.
>
> In the meta-learning setting, we assume that the world model shares some common structure across the task distribution. **The principle behind is that tasks with similar contexts will behave similarly in the world model.** Intuitively, we map tasks into a latent space projected by the world model, and infer task similarity via the world model dynamics. That is, we extrapolate the meta-level knowledge across tasks by the extrapolation ability of the world model, which is more accurate and robust since the world model is intrinsically invariant to behavior policies or collected datasets. Subsequently, we pretrain the world model and then train the meta-policy with a fixed world model. This kind of decoupling assures a more stable process for learning task representations, since the algorithm can solely focus on capturing task similarity via the world model dynamics.
>
>
> ### `Summary Response`
>
> Thank you for your valuable comments, which help us gain more insights into our methodology and motivate promising directions for future work. We are honored to have **your recognition of our work**, and are also grateful that **most reviewers are quite affirmative to our overall contributions**, including the novelty and motivation, the extensiveness of empirical evaluation, and the superior performance.
>
> The corresponding figures of extended experiments can be found in the PDF of Global Response. Please let us know if we have addressed your concerns, and we are more than delighted to have further discussions and improve our manuscript.

---

> > ### Comment · Reviewer_BWEr · 2024-08-11
> > **Response**
> >
> > All of my concerns are well addressed. Thank you. I will increase my rating to accept

---

> > > ### Author Response · Authors · 2024-08-12
> > > **Thank you!**
> > >
> > > Thank you for your time in helping us improve our work. We are more than delighted to see that we could address your all concerns. We sincerely appreciate you raising the score on our work!

---

### Official Review · Reviewer_sbh9 · 2024-07-12

**Soundness:** 3
**Presentation:** 3
**Contribution:** 3
**Rating:** 7
**Confidence:** 3

**Summary:**

The paper introduces a new architecture for Meta-RL based on Decision Transformers. The new architecture uses a world model responsible for efficiently encoding task information from demonstrations. The model also introduces a prompt encoder which acts as a boosting mechanism to the context encoder to enhance the models task adaptation abilities.

**Strengths:**

- Strong experimental evaluation including over many different tasks, settings,f ablations, and comparison against several other techniques
- Strong results in this evaluation, with good demonstration that the introduced components make a difference.
- Simple design with good results
- Good figures and explanations, with std error ranges -- which can be critical for RL

**Weaknesses:**

No major flaws with the paper.
The evaluation covers environments which are relatively homogenous in nature. Performance in MetaWorld is much less differentiating compared to MuJoCo. It would be nice to see more detailed evaluation in non-mujoco tasks.

It would be interesting to see how the technique generalizes across worlds and tasks with more diversity in them, mujoco worlds are extremely minimal with little variation. Something like training on K-levels of an Atari game and generalizing to the remaining N-K levels of well suited Atari games. It is entirely possible the research in this simply isn't ready to tackle hard cases such as this though. Or something like the worlds seen in the Muesli paper.

**Questions:**

Why do you thin the disentanglement works well as an idea? I understand the different perspectives of  adaptation the prompt vs context encoder aim to target, but could you not form the context in manner to do this, for example running it twice with different subsets of experience? Is it the way things are fed in to the prompt vs context encoder (segmenting) and the lack of constraints on the prompt encoders representation?

**Limitations:**

The limitations and future work are well addressed

---

> ### Author Rebuttal · Authors · 2024-08-06
>
> **Q1. It would be nice to see more detailed evaluation in non-mujoco tasks.**
>
> A1. Thank you for your advice. We have conducted new evaluations on Humanoid-Dir and two more Meta-World environments of Sweep and Door-Lock. The following table shows the few-shot testing performance of Meta-DT against baselines using Medium datasets (Figure S2 in Global Response). **The results again demonstrate the consistent superiority of Meta-DT over all baselines in these newly evaluated domains**.
>
> | Environment  | CMT | Prompt-DT | Generalized DT | CORRO | CSRO | Meta-DT |
> | --- | --- | --- | --- | --- | --- | --- |
> | Humanoid-Dir | 548.42 $\pm$ 40.98 | 556.43 $\pm$ 15.84 | 538.95 $\pm$ 7.04 | 429.66 $\pm$ 150.58 | 465.58 $\pm$ 21.78 | **661.08** $\pm$ 57.04 |
> |     Sweep    | 493.06 $\pm$ 72.54 | 593.11 $\pm$ 92.10 | 549.17 $\pm$ 79.35 | 272.67 $\pm$ 77.41 | 446.50 $\pm$ 191.46 | **760.18** $\pm$ 4.87 |
> |  Door-Lock   | 2291.82 $\pm$ 76.88 | 2284.95 $\pm$ 341.20 | 2601.24 $\pm$ 44.42 | 1615.66 $\pm$ 248.00 | 2216.67 $\pm$ 116.84 | **3025.92** $\pm$ 23.41 |
>
>
> **Q2. It would be interesting to see how the technique generalizes across worlds and tasks with more diversity in them, like Atari games.**
>
> A2. Thank you for pointing out this insightful direction. As stated in the limitations section, our generalist model is trained on relatively lightweight datasets compared to popular large models. The urgent trend that RL practitioners are striving to break through is to deploy on significantly large datasets with more diversity, unlocking the scaling law with the transformer architecture. Due to limited time during rebuttal, it is hard for us to conduct systematic empirical evaluation on Atari domains. We will include the above insights into our limitations section, and are looking forward to investigating this promising future direction.
>
>
> **Q3. Why do you think the disentanglement works well as an idea?**
>
> A3. Thank you for your insightful question. Existing context-based or prompt-based methods usually infer task representations by feeding subsets of experience into a trajectory encoder. The offline dataset depends on both the task and the behavior policy. **The task information could be entangled with the features of behavior policies**, thus producing biased task inference at test time due to the shift in behavior policies.
>
> Hence, we attempt to find a stable way to accurately disentangle task-relevant information from behavior policies. In the RL regime, **the world model completely describes the characteristics of the MDP/task, and keeps invariant to behavior policies or collected datasets**. Naturally, it could be a promising alternative for accurately disentangling task beliefs. That is the beginning of motivating the world model disentanglement as our idea.
>
> In the meta-learning setting, we assume that the world model shares some common structure across the task distribution. **The principle behind is that tasks with similar contexts will behave similarly in the world model.** Intuitively, we map tasks into a latent space projected by the world model, and infer task similarity via the world model dynamics. That is, we extrapolate the meta-level knowledge across tasks by the extrapolation ability of the world model, which is more accurate and robust since the world model is intrinsically invariant to behavior policies or collected datasets. Subsequently, we pretrain the world model and then train the meta-policy with a fixed world model. This kind of decoupling assures a more stable process for learning task representations, since the algorithm can solely focus on capturing task similarity via the world model dynamics.
>
> ### `Summary Response`
>
> Thank you for your valuable comments, which help us enhance our experimental evaluation and gain more insightful directions for future work. We are honored to have **your recognition of our work**, and are also grateful that **most reviewers are quite affirmative to our overall contributions**, including the novelty and motivation, the extensiveness of empirical evaluation, and the superior performance.
>
> The corresponding figures of extended experiments can be found in the PDF of Global Response. Please let us know if we have addressed your concerns, and we are more than delighted to have further discussions and improve our manuscript.

---

### Official Review · Reviewer_EjPB · 2024-07-12

**Soundness:** 3
**Presentation:** 2
**Contribution:** 2
**Rating:** 5
**Confidence:** 4

**Summary:**

This work proposes a transformer-based framework for offline meta-reinforcement learning problems. The proposed algorithm utilizes a context-aware world model for task encoding and self-guided prompting. It outperforms existing offline meta-RL baselines.

**Strengths:**

S1. The results in the paper show the performance gain of the proposed method over baselines.

**Weaknesses:**

W1. The work misses the CMT baseline which is cited but not shown as a baseline (https://arxiv.org/abs/2211.08016).

W2. The work lacks novelty as the idea of a world model and Prompt tuning already exists in CMT. Moreover, the transformer architecture is based on a decision transformer.

W3. The run-time is unclear for fare comparison with baselines.

W4. The results for Mujoco benchmark $\texttt{Humanoid-Dir}$ in CSRO are missing. Meta-world results are limited (shown results only for $\texttt{Reach}$), thus hard for a general conclusion.

**Questions:**

Q1. How do the world modeling and prompt tuning in the proposed method differ from CMT?

Q2. How does the method perform in comparison to CMT?

Q3. How does the method perform in $\texttt{Humanoid-Dir}$ and more meta-world benchmarks such as  $\texttt{Hammer}$, $\texttt{Sweep}$, and $\texttt{Door-Lock}$?

Q4. What is the sensitivity with the prompt length $k$?

**Limitations:**

The limitations section is inadequate.

L1. There seems to be high computational overhead during training and inference.

L2. Meta-world benchmarks concluded only from one task which may question the generalization of the proposed method.

---

> ### Author Rebuttal · Authors · 2024-08-06
>
> **Q1. The work misses the CMT baseline which is cited but not shown as a baseline.**
>
> A1. The reason why we did not include CMT as a baseline is that **its code has not been open-sourced**. We emailed CMT's authors for requesting the source code, and received the response that the code would be made public at an uncertain time. Following your suggestion, **we have included CMT as a new baseline** using our own implementation (Figure S1 of Global Response). **The results again demonstrate the consistent superiority of Meta-DT over CMT**.
>
> |Method|Point-Robot|Cheetah-Vel|Cheetah-Dir|Ant-Dir|Hopper-Param|Walker-Param|Reach|
> |---|---|---|---|---|---|---|---|
> |CMT|-12.21$\pm$ 0.51|-131.71$\pm$ 10.51|545.55$\pm$ 7.07|275.67$\pm$ 19.12|325.20$\pm$ 3.62|356.45$\pm$ 5.58|2295.34$\pm$ 121.56|
> |Meta-DT|**-10.18**$\pm$ 0.18|**-99.28**$\pm$ 3.96|**608.18**$\pm$ 4.18|**412.00**$\pm$ 11.53|**348.20**$\pm$ 3.21|**405.12**$\pm$ 11.11|**2458.83**$\pm$ 162.10|
>
> **Q2. How do the world modeling and prompt tuning in the proposed method differ from CMT?**
>
> A2. **We do not tune the prompt**. Meta-DT selects the trajectory segment using the world model to construct the prompt, without tuning any model parameters associated with the prompt. While both involve the concepts of world model and prompt, **Meta-DT differs significantly from CMT in many crucial aspects** including:
>
> - `The way of using the world model for task inference is different.` CMT infers task beliefs at the trajectory level, abstracting a task prompt $z_{\tau}$ from a context trajectory $\tau$ and using that prompt to guide policy generation for the entire episode. In contrast, Meta-DT performs task inference at the transition level, abstracting task representation $z_t$ at each timestep $t$. Hence, Meta-DT can provide more fine-grained guidance to realize high-capacity generalization, and can achieve zero-shot generalization via deriving real-time task inference.
>
> - `The prompt design is quite different.` CMT abstracts task presentation $z_{\tau}$ from a context trajectory and uses $z_{\tau}$ as the task prompt. In contrast, we directly use a trajectory segment $\tau^*$ as the task prompt, enjoying the power of architecture inductive bias like in Prompt-DT. Moreover, CMT tunes the prompt adaptor layer based on relabeling the offline dataset, while Meta-DT does not involve prompt tuning. Our complimentary prompt design is more lightweight and easy-to-implement. By using the world model to help construct the prompt, the world model (algorithmic perspective) and the complementary prompt (architecture perspective) work as an efficient whole to perform accurate task inference for Meta-DT.
>
> - `The pipeline of training the world model and meta-policy is different.` CMT simultaneously learns the world model and meta-policy. In contrast, Meta-DT decouples the learning process, which pretrains the world model and then trains the meta-policy with a fixed world model. This kind of decoupling assures a more stable process for learning task representations, since the algorithm can solely focus on capturing task similarity via the world model dynamics.
>
> - `The algorithm paradigm is different.` Our method is based on decision transformer, a return-conditioned supervised learning paradigm that has achieved promising results on offline RL. It feeds the (return-to-go, state, action) tuple sequence to the causal transformer to predict the action element only. In contrast, CMT uses another paradigm that feeds the (state, action, reward) tuple sequence to the causal transformer to perform autoregressive prediction on all elements.
>
> **Q3. Evaluation on Humanoid-Dir and more Meta-World environments.**
>
> A3. Following your suggestion, **we have conducted new evaluation on Humanoid-Dir and two Meta-World environments of Sweep and Door-Lock**. The following table shows the few-shot testing performance of Meta-DT against extended baselines using Medium datasets (Figure S2 in Global Response). The results again demonstrate the **consistent superiority of Meta-DT** over all baselines in these domains.
>
> |Environment|CMT|Prompt-DT|Generalized DT|CORRO|CSRO|Meta-DT|
> |---|---|---|---|---|---|---|
> |Humanoid-Dir|548.42$\pm$ 40.98|556.43$\pm$ 15.84|538.95$\pm$ 7.04 |429.66$\pm$ 150.58|465.58$\pm$ 21.78|**661.08**$\pm$ 57.04|
> |Sweep|493.06$\pm$ 72.54|593.11$\pm$ 92.10|549.17$\pm$ 79.35|272.67$\pm$ 77.41|446.50$\pm$ 191.46|**760.18**$\pm$ 4.87|
> |Door-Lock|2291.82$\pm$ 76.88|2284.95$\pm$ 341.20|2601.24$\pm$ 44.42 |1615.66$\pm$ 248.00|2216.67$\pm$ 116.84|**3025.92**$\pm$ 23.41|
>
> **Q4. The run-time during training and inference is unclear for fare comparison with baselines.**
>
> A4. Please refer to our `A1 to Reviewer iZYy` for the tables of model parameters, training time, and inference time for one episode. Compared to DT-based baselines, our methods incurrs about 15\% increase in model parameters, and about 10\% increase in training and inference time. **Considering the significant performance gain of our method, this lightweight cost is likely acceptable.** Moreover, Meta-DT's computation cost is even lower than other baselines.
>
> **Q5. What is the sensitivity with the prompt length $k$?**
>
> A5. We have conducted new experiments to investigate the influence of prompt length $k$ on Meta-DT’s performance (Figure S3 in Global Response). Generally, **the performance of Meta-DT is not sensitive to the pre-defined value of prompt length**.
>
> |Prompt length $k$|3|5|7|
> |---|---|---|---|
> |Point-Robot|**-10.18**$\pm$ 0.18|-10.19$\pm$ 0.15|-10.22$\pm$ 0.04|
> |Cheetah-Dir|606.69$\pm$ 2.92|**608.18**$\pm$ 4.18|605.09$\pm$ 6.31|
> |Ant-Dir|408.97$\pm$ 18.01|412.00$\pm$ 11.53|**413.09**$\pm$ 8.82|
>
> **Q6. The limitations section is inadequate.**
>
> A6. We have discussed three threads of our limitations. During rebuttal, the valuable comments from all reviewers also give us new insights into our method, and we will have a more thorough discussion on limitations based on these comments.

---

> ### Author Response · Authors · 2024-08-06
> **Response Summary by Authors**
>
> Thank you for your valuable review comments, which help us gain more critical insights on the difference from existing works, and further enhance our empirical evaluation. We summarize the three main concerns as
>
> - `The difference from CMT.` While both involve the concepts of world model and prompt, **Meta-DT differs significantly from CMT in many crucial aspects** (refer to our A2). Also, **other four reviewers** (we refer to iZYy as R1, LRM6 as R2, sbh9 as R4, and BWEr as R5) **are quite affirmative to our novelty and motivation** ("a novel Meta-DT" by R1, "a clear motivation... quite novel and inspiring" by R2, "a new architecture... no major flaws with the paper" by R4, and "a new framework" by R5). We hope that our novelty and distinct contributions have been adequately justified.
>
> - `Empirical comparison to CMT.`We did not include CMT as a baseline since we could not get its source code. Following your advice, **we have added CMT as a new baseline**, and evaluation results again show the superiority of our method.
>
> - `Evaluation on more environments.` We evaluated our method on **seven environments adopted from three classical benchmarks** in meta-RL. Also, **other four reviewers are quite affirmative to the extensiveness of our empirical evaluation** ("shows the good performance" by R1, "comprehensive experiments on multiple benchmarks" by R2, "strong experimental evaluation over many different tasks" by R4, and "comprehensive experimental results" by R5). Following your advice, **we have conducted new evaluations on Humanoid-Dir and two more Meta-World environments**. Finally, results of **a total of 10 environments from 3 benchmarks** again verify our advantages. We hope that our extended experiments can further verify the generalization of our method.
>
> The corresponding figures of extended experiments can be found in Global Response. Again, we appreciate your time in reviewing our manuscript and your valuable comments. We would like to justify our method to you in more detail, and we are more than delighted to have further discussions to improve our manuscript. If our response has addressed your concerns, we would be grateful if you could re-evaluate our work.

---

> > ### Comment · Reviewer_EjPB · 2024-08-11
> >
> > Thank you for the detailed response. As the majority of my concerns have been addressed, I will increase my rating.

---

> > > ### Author Response · Authors · 2024-08-12
> > > **Thank you!**
> > >
> > > Thank you for your time in helping us improve our work. We are happy to see that we could address your main concerns. Thank you sincerely for raising the score on our work. We truly appreciate it!

---

### Official Review · Reviewer_LRM6 · 2024-07-13

**Soundness:** 3
**Presentation:** 3
**Contribution:** 3
**Rating:** 6
**Confidence:** 3

**Summary:**

This paper proposes a novel meta-RL framework called Meta Decision Transformer (Meta-DT). It leverages robust task representation learning via world model disentanglement to achieve task generalization in offline meta-RL. Firstly, it pretrains a context-ware world model to capture the task-relevant information from the offline dataset. Then,  it guides the sequence generation of decision transformer using the task representation and the self-guided prompt from past trajectories. The authors conduct extensive experiments on multiple benchmarks. Meta-DT demonstrates better few- and zero-shot generalization ability than other baselines.

**Strengths:**

1. This paper has a clear motivation in disentangling the task specific information. It is quite novel and inspiring to make use of the prompt to provide task-specific context and exploit architecture inductive bias.

2. The authors conduct comprehensive experiments on multiple benchmarks, showing notable performance gains in many tasks of both few-shot and zero-shot scenarios. The ablation studies are also designed properly to justify the algorithm design.

3. The paper is well-organized and easy to understand. The authors also provide detailed pseudo-codes and hyper-parameters in the appendix for reproduction.

**Weaknesses:**

1. From my perspective, the generalization ability of meta-DT depends heavily on the extrapolation ability of the context encoder. Although it is intuitive for some continuous tasks like Cheetah-Vel, this extrapolation tends to be quite hard for some other discrete tasks like Cheetah-Dir. Despite the good experiment performance, some intuitive explanation and theoretical justification are also desirable.

2.  To evaluate the generalization ability more comprehensively, it may be helpful to provide the experiments with different number of training tasks. It is also interesting to see the extrapolation ability limit of meta-DT. For example, what if the training tasks of Ant-Dir are sampled from U[0,1.5$\pi$] with hold-out tasks are from U[1.5$\pi$, 2$\pi$].

**Questions:**

Please also consider responding to the Weaknesses.

**Limitations:**

The authors have discussed the limitations.

---

> ### Author Rebuttal · Authors · 2024-08-06
>
> **Q1. Some intuitive explanation on the extrapolation ability are also desirable.**
>
> A1. Thank you for your insightful comments. Meta-DT's generalization comes from the extrapolation ability of the context-aware world model $W(r,s'|s,a; z^i)$. The world model completely describes the characteristics of the MDP/task. In the meta-learning setting, we assume that the world model shares some common structure across the task distribution, with a latent representation $z^i$ to approximate the unknown context of task $i$. **The principle behind is that tasks with similar contexts will behave similarly in the world model.** Intuitively, we map tasks into a latent space projected by the world model, and infer task similarity via the world model dynamics. That is, we extrapolate the meta-level knowledge across tasks by the extrapolation ability of the world model, which is more accurate and robust since the world model is intrinsically invariant to behavior policies or collected datasets. For discrete tasks like Cheetah-Dir, the world model also shares some common structure across the task distribution, e.g., the kinematics principle or locomotion skills. Hence, **the extrapolation of the world model works for both continuous and discrete tasks**. We will enhance the intuitive explanation of our method in the revised paper.
>
>
> **Q2. It may be helpful to provide the experiments with different numbers of training tasks.**
>
> A2. Following your advice, we have conducted new experiments to evaluate Meta-DT with a different number of training tasks. The following table shows the few-shot test performance of Meta-DT using Medium datasets (Figure S4 in Global Response). **Increasing the number of training tasks can usually improve the generalization ability to test tasks, especially in harder tasks like Ant-Dir**.
>
> The result also matches the intuition of function approximation in machine learning. The "generalization" refers to the question: How can experience with a limited subset of the data space be usefully generalized to produce a good approximation over a much larger subset? In the single-task setting, increasing the valid sample points (before being overfitted) can generally improve the function approximation of the sample space and the generalization to unseen samples at testing. In the meta-learning setting, increasing the valid task points (before being overfitted) can usually boost the function approximation of the task space and the generalization to unseen tasks at testing. We will include this new experiment and the interesting findings in our revised paper.
>
> | Number of training tasks | 15 | 30 | 45 |
> | --- | --- | --- | --- |
> | Point-Robot | -10.20 $\pm$ 0.28 | -10.45 $\pm$ 0.22 | **-10.18** $\pm$ 0.18 |
> | Cheetah-Vel | **-85.42** $\pm$ 1.59 | -95.14 $\pm$ 0.68 | -99.28 $\pm$ 3.96 |
> |   Ant-Dir   | 388.50 $\pm$ 9.02 | 402.71 $\pm$ 5.00 | **412.00** $\pm$ 11.53 |
>
> **Q3. It is also interesting to see the extrapolation ability limit of meta-DT.**
>
> A3. Following your suggestion, we have conducted new experiments on Ant-Dir to evaluate Meta-DT's generalization ability to out-of-distribution (OOD) tasks. The following table shows the few-shot performance of Meta-DT with OOD test tasks (Figure S5 in Global Response). In this case, we sample training tasks with a goal direction of $U[0, 1.5\pi]$, and then test on tasks of $U[1.5\pi, 2\pi]$.
>
> Obviously, **Meta-DT can still obtain better performance than baselines on OOD test tasks**, which again verifies our superiority. As stated in our A1 to Q1, we extrapolate the meta-level knowledge across tasks by the extrapolation ability of the world model, which is more accurate and robust since the world model is intrinsically invariant to behavior policies or collected datasets. For tasks like Ant-Dir, the world model shares some common structure across the task distribution (even for OOD tasks), e.g., the kinematics principle or locomotion skills. Hence, the extrapolation of the world model also works for OOD test tasks in this case. We will include this interesting experiment in the revised paper.
>
> | OOD Testing  | CMT | Prompt-DT | Generalized DT | CORRO | CSRO | Meta-DT |
> | --- | --- | --- | --- | --- | --- | --- |
> | Ant-Dir | 271.74 $\pm$ 102.81 | 281.73 $\pm$ 12.29 | 148.83 $\pm$ 6.06 | 141.79 $\pm$ 94.20 | 208.92 $\pm$ 15.47 | **412.67** $\pm$ 5.15 |
>
> ### `Summary Response`
>
> Thank you for your valuable suggestions, which help us gain more critical insights into the extrapolation ability of Meta-DT, and further enhance our experimental evaluation. The corresponding figures of extended experiments can be found in the PDF of Global Response. We are honored to have **your recognition of our method**, and are also grateful that **most reviewers are quite affirmative to our overall contributions**, including the novelty and motivation, the extensiveness of empirical evaluation, and the superior performance.
>
> Please let us know if we have addressed your concerns. We are more than delighted to have further discussions and improve our manuscript.

---

> ### Comment · Reviewer_LRM6 · 2024-08-11
>
> Thank you for the detailed response. This has helped to clarify my questions. I recognize the contribution of this work, and I would like to keep my original rating to vote for acceptance.

---

> > ### Author Response · Authors · 2024-08-11
> > **Thank you!**
> >
> > Thank you for your time in helping us improve our work. We are happy to see that we could address your concerns. We sincerely appreciate your recognition of our contribution and your vote to accept our work!

---

### Official Review · Reviewer_iZYy · 2024-07-15

**Soundness:** 3
**Presentation:** 3
**Contribution:** 3
**Rating:** 6
**Confidence:** 3

**Summary:**

This paper proposes a novel Meta-DT method that leverages the task representation from the world model disentanglement. Compared with the previous works, the expert demonstration is not necessary. This method could get the task representation from the trained encoder which is used as the guidance for the autoregressive training. This paper also incorporates an additional prompt as the complementary. The empirical study shows the good performance of the Meta-DT compared with baselines.

**Strengths:**

- This paper proposes a novel Meta-DT that disentangles the task information from the trajectories as the prompt in Meta RL tasks.
- This paper could outperform the baselines in most of the settings, especially on low-quality demonstration sets.

**Weaknesses:**

- This paper needs to train a world model which is used to represent the dynamic information for the task. This method is not as light as the previous work like Prompt-DT.
- This paper shows the sub-performance compared with prompt DT when the demonstration set is expert.

**Questions:**

- Can you explain more about the training hours and parameters for this method?
- Can you explain the experiments why the method does not outperform the baselines in the expert demonstration?

**Limitations:**

See weaknesses and questions.

---

> ### Author Rebuttal · Authors · 2024-08-06
>
> **Q1. Explain more about the training hours and parameters for this method.**
>
> A1. The following tables show the number of model parameters, the training time, and inference time for one episode. Compared to DT-based baselines, our method introduces a lightweight world model that consists of about 15\% parameters of the DT backbone. Thanks to this lightweight design, Meta-DT incurrs about 10\% runtime increase during training and inference. **Considering the significant performance gain of our method, this lightweight cost is likely acceptable.** Moreover, Meta-DT's computation cost is even lower than other baselines.
>
> |`Model Size`|CMT|Prompt-DT|Generalized DT|CORRO|CSRO|Meta-DT|
> |---|---|---|---|---|---|---|
> |Point-Robot|675,989|603,259|600,754|705,961|858,480|702,758|
> |Cheetah-Vel|730,891|657,819|629,718|740,273|895,448|755,996|
> |Cheetah-Dir|730,891|657,819|629,718|740,273|895,448|755,996|
> |Ant-Dir|734,500|661,284|631,240|754,357|910,527|758,965|
> |Hopper-Param|726,079|653,199|627,651|756,827|885,014|751,274|
> |Walker-Param|729,720|658,872|629,286|735,665|893,808|755,129|
>
> |`Training time (s)`|CMT|Prompt-DT|Generalized DT|CORRO|CSRO|Meta-DT|
> |---|---|---|---|---|---|---|
> |Point-Robot|874|762|594|1,315|980|824|
> |Cheetah-Vel|1,068|758|600|1,609|1,076|1,050|
> |Cheetah-Dir|912|790|596|1,599|1,052|877|
> |Ant-Dir|5,076|4,605|3,021|9,880|6,179|4,930|
> |Hopper-Param|1,305|770|596|2,113|1,073|1,018|
> |Walker-Param|1,304|760|595|2,211|1,067|1,027|
>
> |`Inference time (ms)`|CMT|Prompt-DT|Generalized DT|CORRO|CSRO|Meta-DT|
> |---|---|---|---|---|---|---|
> |Point-Robot|9.52|8.68|46|10|15|9.16|
> |Cheetah-Vel|50.44|45.84|468|125|168|48.92|
> |Cheetah-Dir|50.40|48.2|467|123|167|48.90|
> |Ant-Dir|55.72|50.11|518|162|244|53.24|
> |Hopper-Param|51.77|48.64|488|146|210|51.00|
> |Walker-Param|54.13|49.60|516|147|231|53.01|
>
> **Q2. Explain the experiments why the method does not outperform the baselines in the expert demonstration.**
>
> A2. In experiments with Expert datasets as shown in Table 4 and Figure 10, Meta-DT outperforms three baselines (Generalized DT, CORRO, and CSRO) to a large extent in all environments.
>
> Compared to Prompt-DT, Meta-DT obtains **significantly better performance** in Point-Robot (-6.90 vs. -7.99), Cheetah-Vel (-52.42 vs. -133.78), and Ant-Dir (961.27 vs. 678.07), and obtains **extremely close performance** in Cheetah-Dir (874.91 vs. 960.32), Hopper-Param (383.51 vs. 393.79), and Walker-Param (437.79 vs. 449.15). **Taken together, it can be empirically verified that Meta-DT still outperforms Prompt-DT when evaluated on Expert datasets.**
>
> With Expert datasets, Prompt-DT can access high-quality expert demonstrations as the task prompt to guide policy generation. Within the expert data, the behavior policy exactly equals the optimal policy and totally aligns with task characteristics, thus it can capture task-relevant information very well. Hence, Prompt-DT can achieve the same level of performance as Meta-DT in several environments. When faced with inferior datasets, the task information could be entangled with the features of behavior policies, producing biased task inference at test time. Hence, the performance of Prompt-DT might drop a lot, while Meta-DT can still obtain satisfactory performance due to accurately capturing task information via world model disentanglement.
>
>
> **Q3. This method is not as light as the previous work like Prompt-DT.**
>
> A3. We agree with you on this point, as our method introduces a pretrained context-aware world model. In contrast, Prompt-DT uses the same network architecture as DT, with some expert demonstrations prepended to the DT's input.
>
> Our motivation of introducing the additional world model is to disentangle task-relevant information from behavior policies, thus more accurately inferring task beliefs. With this effective disentanglement, our method is verified to be robust to the dataset quality and is more practical with fewer prerequisites in real-world scenarios. This is what Prompt-DT lacks, since it is sensitive to the quality of prompt demonstrations and can achieve satisfactory performance only when expert demonstrations are available at test time.
>
>
> ### `Summary Response`
>
> Thank you for your valuable review comments, which help us gain more insights on comparison to existing works like Prompt-DT, and further enhance our experimental illustration. We are honored to have **your recognition on our method, especially on its novelty** ("a novel Meta-DT method") **and superior performance** ("shows the good performance... outperform the baselines").
>
> Also, we are grateful that **most reviewers** (we refer to LRM6 as R2, EjPB as R3, sbh9 as R4, and BWEr as R5) **are quite affirmative to our overall contributions, including the novelty and motivation** ("a clear motivation... quite novel and inspiring" by R2, "a new architecture... no major flaws with the paper" by R4, and "a new framework" by R5), **the extensiveness of empirical evaluation** ("comprehensive experiments on multiple benchmarks" by R2, "strong experimental evaluation over many different tasks" by R4, and "comprehensive experimental results" by R5), and **the superior performance** ("showing notable performance gains" by R2, "show the performance gain" by R3, "strong results in this evaluation" by R4, and "achieves superior generalization... demonstrating its practical effectiveness" by R5).
>
> Please let us know if we have addressed your concerns. We are more than delighted to have further discussions and improve our manuscript. If our response has addressed your concerns, we would be grateful if you could re-evaluate our work.

---

> ### Author Response · Authors · 2024-08-12
> **Looking forward to further discussions!**
>
> Dear Reviewer,
>
> Thank you for your insightful comments. We were wondering if our response and revision have resolved your concerns. We have addressed your initial questions through our rebuttal and are eager to clarify any further points you might raise. Please feel free to provide additional feedback. We greatly appreciate your continued engagement.
>
> Best regards,
>
> Authors

---

> > ### Comment · Reviewer_iZYy · 2024-08-13
> >
> > Thanks for the detailed explanation and additional experiments. I will raise my score.

---

> > > ### Author Response · Authors · 2024-08-13
> > > **Thank you!**
> > >
> > > Thank you very much for your constructive feedback and your time in helping us improve our work. We sincerely appreciate you raising the score on our work!

---

### Author Rebuttal · Authors · 2024-08-06

# Revision Summary

We thank the reviewers for their valuable feedback (we refer to iZYy as R1, LRM6 as R2, EjPB as R3, sbh9 as R4, and BWEr as R5). We are grateful that **most reviewers are quite affirmative to our overall contributions**, including **the novelty and motivation** ("a novel Meta-DT" by R1, "a clear motivation... quite novel and inspiring” by R2, "a new architecture... no major flaws with the paper" by R4, and "a new framework" by R5), **the extensiveness of empirical evaluation** ("comprehensive experiments on multiple benchmarks" by R2, "strong experimental evaluation over many different tasks" by R4, and "comprehensive experimental results" by R5), and **the superior performance** ("shows the good performance" by R1, "showing notable performance gains" by R2, "show the performance gain" by R3, "strong results in this evaluation" by R4, and "achieves superior generalization... demonstrating its practical effectiveness" by R5).

We have made **a number of changes** to address all reviewers' suggestions and concerns. A short summary of the modifications is made as

1. We include CMT as a new baseline for experimental evaluation.
2. We conduct new empirical evaluation on Humanoid-Dir and two Meta-World environments of Sweep and Door-Lock.
3. We conduct new hyperparameter analysis experiments to investigate the influence of prompt length $k$ on Meta-DT’s performance.
4. We conduct new experiments to demonstrate Meta-DT's performance with a different number of training tasks.
5. We include new experiments to evaluate Meta-DT's generalization ability to out-of-distribution tasks.
6. We show and analyze the number of model parameters, the training time, and the inference time for Meta-DT and baselines.
7. We further justify our methodology including: i) the significant differences from CMT and corresponding superiorities over CMT, ii) the motivation and benefits of world model disentanglement, and iii) limitations of our method, challenges in extending to harder cases, and promising future directions.

In summary, we have **significantly extended our empirical evaluation** based on the comprehensive experimental results in our original manuscript. Impressively, **the results of massively extended experiments are generally consistent with observations and conclusions from our original manuscript**. The tabular results are given in responses to each reviewer, and corresponding figures can be found in the PDF of Global Response.

Please let us know if we have addressed your concerns. We are more than delighted to have further discussions and improve our manuscript. If our response has addressed your concerns, we would be grateful if you could re-evaluate our work.

---

### Decision · Program_Chairs · 2024-09-25

**Decision:**

Accept (poster)

**Comment:**

All reviewers were positive about the paper after the rebuttal. Some reviewers especially praised the paper's simplicity and strong generalization results. AC liked the experiments with varying number of tasks which helped reveal the method's potential and limitation in generalization.